# Lipschitz Bandits with Batched Feedback

**Yasong Feng**
Shanghai Center for Mathematical Sciences
Fudan University
ysfeng20@fudan.edu.cn

**Zengfeng Huang**[*]
School of Data Science
Fudan University
huangzf@fudan.edu.cn

**Tianyu Wang**[*]
Shanghai Center for Mathematical Sciences
Fudan University
wangtianyu@fudan.edu.cn

## Abstract

In this paper, we study Lipschitz bandit problems with batched feedback, where the expected reward is Lipschitz and the reward observations are communicated to the player in batches. We introduce a novel landscape-aware algorithm, called Batched Lipschitz Narrowing (BLiN), that optimally solves this problem. Specifically, we show that for a $T$-step problem with Lipschitz reward of zooming dimension $d_z$, our algorithm achieves theoretically optimal (up to logarithmic factors) regret rate $\widetilde{\mathcal{O}}\left(T^{\frac{d_z+1}{d_z+2}}\right)$ using only $\mathcal{O}\left(\log\log T\right)$ batches. We also provide complexity analysis for this problem. Our theoretical lower bound implies that $\Omega(\log\log T)$ batches are necessary for any algorithm to achieve the optimal regret. Thus, BLiN achieves optimal regret rate using minimal communication.

## 1 Introduction

Multi-Armed Bandit (MAB) algorithms aim to exploit the good options while explore the decision space. These algorithms and methodologies find successful applications in artificial intelligence and reinforcement learning [e.g., 37]. While the classic MAB setting assumes that the rewards are immediately observed after each arm pull, real-world data often arrives in different patterns. For example, observations from clinical trials are often be collected in a batched fashion [34]. Another example is from online advertising, where strategies are tested on multiple customers at the same time [10]. In such cases, any observation-dependent decision-making should comply with this data-arriving pattern, including MAB algorithms.

In this paper, we study the Lipschitz bandit problem with batched feedback – a MAB problem where the expected reward is Lipschitz and the reward observations are communicated to the player in batches. In such settings, rewards are communicated only at the end of the batches, and the algorithm can only make decisions based on information up to the previous batch. Existing Lipschitz bandit algorithms heavily rely on timely access to the reward samples, since the partition of arm space may change at any time. Therefore, they can not solve the batched feedback setting. To address this difficulty, we present a novel adaptive algorithm for Lipschitz bandits with communication constraints, named *Batched Lipschitz Narrowing* (BLiN). BLiN learns the landscape of the reward by adaptively narrowing the arm set, so that regions of high reward are more frequently played. Also, BLiN determines the data collection procedure adaptively, so that only very few data communications are needed.

---

[*]Corresponding authors

36th Conference on Neural Information Processing Systems (NeurIPS 2022).

The above BLiN procedure achieves optimal regret rate $\widetilde{\mathcal{O}}\left(T^{\frac{d_z+1}{d_z+2}}\right)$ ($d_z$ is the zooming dimension [26, 15]), and can be implemented in a clean and easy-to-implement form. In addition to achieving the optimal regret rate, BLiN is also optimal in the following senses:

- BLiN's communication complexity is optimal. BLiN only needs $\mathcal{O}(\log \log T)$ rounds of communications to achieve the optimal regret rate (Theorem 2), and no algorithm can achieve this rate with fewer than $\Omega(\log \log T)$ rounds of communications (Corollary 2).
- BLiN's time complexity is optimal (Remark 1): if the arithmetic operations and sampling are of complexity $\mathcal{O}(1)$, then the time complexity of BLiN is $\mathcal{O}(T)$, which improve the best known time complexity $\mathcal{O}(T \log T)$ for Lipschitz bandit problems [15].

## 1.1 Settings & Preliminaries

For a Lipschitz bandit problem (with communication constraints), the arm set is a compact doubling metric space $(\mathcal{A}, d_{\mathcal{A}})$. The expected reward $\mu : \mathcal{A} \to \mathbb{R}$ is 1-Lipschitz with respect to the metric $d_{\mathcal{A}}$, that is, $|\mu(x_1) - \mu(x_2)| \le d_{\mathcal{A}}(x_1, x_2)$ for any $x_1, x_2 \in \mathcal{A}$.

At time $t \le T$, the learning agent pulls an arm $x_t \in \mathcal{A}$ that yields a reward sample $y_t = \mu(x_t) + \epsilon_t$, where $\epsilon_t$ is a mean-zero independent sub-Gaussian noise. Without loss of generality, we assume that $\epsilon_t \sim \mathcal{N}(0, 1)$, since generalizations to other sub-Gaussian noises are not hard.

Similar to most bandit learning problems, the agent seeks to minimize regret in the batched feedback environment. The regret is defined as $R(T) = \sum_{t=1}^{T} (\mu^* - \mu(x_t))$, where $\mu^*$ denotes $\max_{x \in \mathcal{A}} \mu(x)$. For simplicity, we define $\Delta_x = \mu^* - \mu(x)$ (called optimality gap of $x$) for all $x \in \mathcal{A}$.

### 1.1.1 Doubling Metric Spaces and the $([0,1]^d, \|\cdot\|_\infty)$ Metric Space

By the Assouad's embedding theorem [6], the (compact) doubling metric space $(\mathcal{A}, d_{\mathcal{A}})$ can be embedded into a Euclidean space with some distortion of the metric; See [43] for more discussions in a machine learning context. Due to existence of such embedding, the metric space $([0,1]^d, \|\cdot\|_\infty)$, where metric balls are hypercubes, is sufficient for the purpose of our paper. For the rest of the paper, we will use hypercubes in algorithm design for simplicity, while our algorithmic idea generalizes to other doubling metric spaces.

### 1.1.2 Zooming Numbers and Zooming Dimensions

An important concept for bandit problems in metric spaces is the zooming number and the zooming dimension [26, 14, 39], which we discuss now.

Define the set of $r$-optimal arms as $S(r) = \{x \in \mathcal{A} : \Delta_x \le r\}$. For any $r = 2^{-i}$, the decision space $[0,1]^d$ can be equally divided into $2^{di}$ cubes with edge length $r$, which we call *standard cubes* (also referred to as dyadic cubes). The $r$-zooming number is defined as

$$N_r := \#\{C : C \text{ is a standard cube with edge length } r \text{ and } C \subset S(16r)\}.$$

In words, $N_r$ is the $r$-packing number of the set $S(16r)$ in terms of standard cubes. The zooming dimension is then defined as

$$d_z := \min\{d \ge 0 : \exists a > 0, \ N_r \le ar^{-d}, \ \forall r = 2^{-i} \text{ for some } i \in \mathbb{N}\}.$$

Moreover, we define the zooming constant $C_z$ as the minimal $a$ to make the above inequality true for $d_z$, $C_z = \min\{a > 0 : \ N_r \le ar^{-d_z}, \ \forall r = 2^{-i} \text{ for some } i \in \mathbb{N}\}$.

Zooming dimension $d_z$ can be significantly smaller than ambient dimension $d$ and can be zero. For a simple example, consider a problem with ambient dimension $d = 1$ and expected reward function $\mu(x) = x$ for $0 \le x \le 1$. Then for any $r = 2^{-i}$ with $i \ge 4$, we have $S(16r) = [1 - 16r, 1]$ and $N_r = 16$. Therefore, for this problem the zooming dimension equals to 0, with zooming constant $C_z = 16$.

## 1.2 Batched feedback pattern and our results

In the batched feedback setting, for a $T$-step game, the player determines a grid $\mathcal{T} = \{t_0, \cdots, t_B\}$ *adaptively*, where $0 = t_0 < t_1 < \cdots < t_B = T$ and $B \ll T$. During the game, reward observations

are communicated to the player only at the grid points $t_1, \cdots, t_B$. As a consequence, for any time $t$ in the $j$-th batch, that is, $t_{j-1} < t \le t_j$, the reward $y_t$ cannot be observed until time $t_j$, and the decision made at time $t$ depends only on rewards up to time $t_{j-1}$. The determination of the grid $\mathcal{T}$ is adaptive in the sense that the player chooses each grid point $t_j \in \mathcal{T}$ based on the operations and observations up to the previous point $t_{j-1}$.

In this work, we present BLiN algorithm to solve Lipschitz bandits under batched feedback. During the learning procedure, BLiN detects and eliminates the 'bad area' of the arm set in batches and partition the remaining area according to an appropriate *edge-length sequence*. Our first theoretical upper bound is that with simple Doubling Edge-length Sequence $r_m = 2^{-m+1}$, BLiN achieves optimal regret rate $\widetilde{\mathcal{O}}\left(T^{\frac{d_z+1}{d_z+2}}\right)$ by using $\mathcal{O}(\log T)$ batches.

**Theorem 1.** *With probability exceeding $1 - \frac{2}{T^6}$, the $T$-step total regret $R(T)$ of BLiN with Doubling Edge-length Sequence (D-BLiN) satisfies*

$$R(T) \lesssim T^{\frac{d_z+1}{d_z+2}} \cdot (\log T)^{\frac{1}{d_z+2}},$$

*where $d_z$ is the zooming dimension of the problem instance. In addition, D-BLiN only needs no more than $\mathcal{O}(\log T)$ rounds of communications to achieve this regret rate. Here and henceforth, $\lesssim$ only omits constants.*

While D-BLiN is efficient for batched Lipschitz bandits, its communication complexity is not optimal. We then propose a new edge-length sequence, which we call Appropriate Combined Edge-length Sequence (ACE Sequence) to improve the algorithm. The idea behind this sequence is that by appropriately combining some batches, the algorithm can achieve better communication bound without incurring increased regret. As we shall see, BLiN with ACE Sequence (A-BLiN) achieves regret rate $\widetilde{\mathcal{O}}\left(T^{\frac{d_z+1}{d_z+2}}\right)$ with only $\mathcal{O}(\log \log T)$ batches.

**Theorem 2.** *With probability exceeding $1 - \frac{2}{T^6}$, the $T$-step total regret $R(T)$ of A-BLiN satisfies*

$$R(T) \lesssim T^{\frac{d_z+1}{d_z+2}} \cdot (\log T)^{\frac{1}{d_z+2}} \cdot \log \log T,$$

*where $d_z$ is the zooming dimension of the problem instance. In addition, Algorithm 1 only needs no more than $\mathcal{O}(\log \log T)$ rounds of communications to achieve this regret rate.*

As a comparison, seminal works [26, 39, 15] show that the optimal regret bound for Lipschitz bandits without communications constraints, where the reward observations are immediately observable after each arm pull, is $R(T) \lesssim T^{\frac{d_z+1}{d_z+2}} \cdot (\log T)^{\frac{1}{d_z+2}}$ in terms of zooming dimension, and

$$R(T) \lesssim R_z(T) \triangleq \inf_{r_0} \left\{ r_0 T + \sum_{r=2^{-i}, r \ge r_0} \frac{N_r}{r} \log T \right\} \tag{1}$$

in terms of zooming number. Therefore, A-BLiN achieves optimal regret rate of Lipschitz bandits by using very few batches.

Moreover, our lower-bound analysis shows that $\Omega(\log \log T)$ batches are necessary for any algorithm to achieve the optimal regret rate. Thus, BLiN is optimal in terms of both regret and communication.

## 1.3 Related Works

The history of the Multi-Armed Bandit (MAB) problem can date back to Thompson [42]. Solvers for this problems include the UCB algorithms [28, 4, 7], the arm elimination method [20, 32, 36], the $\epsilon$-greedy strategy [7, 40], the exponential weights and mirror descent framework [8].

Recently, with the prevalence of distributed computing and large-scale field experiments, the setting of batched feedback has captured attention [e.g., 17]. Perchet et al. [33] mainly consider batched bandit with two arms, and a matching lower bound for static grid is proved. It was then generalized by Gao et al. [21] to finite-armed bandit problems. In their work, the authors designed an elimination method for finite-armed bandit problem and proved matching lower bounds for both static and adaptive grid. Soon afterwards, Zhang et al. [46] studies inference for batched bandits. Esfandiari et al. [19] studies batched linear bandits and batched adversarial bandits. Han et al. [22] and Ruan et

al. [35] provide solutions for batched contextual linear bandits. Li and Scarlett [29] studies batched Gaussian process bandits. Batched dueling bandits have also been studied by [2]. Parallel to the regret control regime, best arm identification with limited number of batches was studied in [1] and [23]. Top-$k$ arm identification in the collaborative learning framework is also closely related to the batched setting, where the goal is to minimize the number of iterations (or communication steps) between agents. In this setting, tight bounds have been obtained in the recent works [41, 24]. Yet the problem of Lipschitz bandit with communication constraints remains unsolved.

The Lipschitz bandit problem is important in its own stand. The Lipschitz bandit problem was introduced as "continuum-armed bandits" [3], where the arm space is a compact interval. Along this line, bandits that are Lipschitz (or Hölder) continuous have been studied. For this problem, Kleinberg [25] proves a $\Omega(T^{2/3})$ lower bound and introduced a matching algorithm. Under extra conditions on top of Lipschitzness, regret rate of $\widetilde{\mathcal{O}}(T^{1/2})$ was achieved [9, 18]. For general (doubling) metric spaces, the Zooming bandit algorithm [26] and the Hierarchical Optimistic Optimization (HOO) algorithm [15] were developed. In more recent years, some attention has been focused on Lipschitz bandit problems with certain extra structures. To name a few, Bubeck et al. [16] study Lipschitz bandits for differentiable rewards, which enables algorithms to run without explicitly knowing the Lipschitz constants. Wang et al. [44] studied discretization-based Lipschitz bandit algorithms from a Gaussian process perspective. Magureanu et al. [31] derive a new concentration inequality and study discrete Lipschitz bandits. The idea of robust mean estimators [11, 5, 13] was applied to the Lipschitz bandit problem to cope with heavy-tail rewards, leading to the development of a near-optimal algorithm for Lipschitz bandit with heavy-tailed rewards [30]. Lipschitz bandits where a clustering is used to infer the underlying metric, has been studied by [45]. Contextual Lipschitz bandits have also been studied by [39] and [27]. Yet all of the existing works for Lipschitz bandits assume that the reward sample is immediately observed after each arm pull, and none of them solve the Lipschitz bandit problem with communication constraints.

This paper is organized as follows. In section 2, we introduce the BLiN algorithm and give a visual illustration of the algorithm procedure. In section 3, we prove that BLiN with ACE Sequence achieves the optimal regret rate using only $\mathcal{O}\left(\log\log T\right)$ rounds of communications. Section 4 provides information-theoretical lower bounds for Lipschitz bandits with communication constraints, which shows that BLiN is optimal in terms of both regret and rounds of communications. Experimental results are presented in Section 5.

## 2 Algorithm

With communication constraints, the agent's knowledge about the environment does not accumulate within each batch. This characteristic of the problem suggests a 'uniform' type algorithm – we shall treat each step within the same batch equally. Following this intuition, in each batch, we uniformly play the remaining arms, and then eliminate arms of low reward after the observations are communicated. Next we describe the uniform play rule and the arm elimination rule.

**Uniform Play Rule:** At the beginning of each batch $m$, a collection of subsets of the arm space $\mathcal{A}_m = \{C_{m,1}, C_{m,2}, \cdots, C_{m,|\mathcal{A}_m|}\}$ is constructed. This collection of subset $\mathcal{A}_m$ consists of standard cubes, and all cubes in $\mathcal{A}_m$ have the same edge length $r_m$. We will detail the construction of $\mathcal{A}_m$ when we describe the arm elimination rule. We refer to cubes in $\mathcal{A}_m$ as active cubes of batch $m$.

During batch $m$, each cube in $\mathcal{A}_m$ is played $n_m := \frac{16 \log T}{r_m^2}$ times, where $T$ is the total time horizon. More specifically, within each $C \in \mathcal{A}_m$, arms $x_{C,1}, x_{C,2}, \cdots, x_{C,n_m} \in C$ are played.[2] The reward samples $\{y_{C,1}, y_{C,2}, \cdots, y_{C,n_m}\}_{C \in \mathcal{A}_m}$ corresponding to $\{x_{C,1}, x_{C,2}, \cdots, x_{C,n_m}\}_{C \in \mathcal{A}_m}$ will be collected at the end of the this batch.

**Arm Elimination Rule:** At the end of batch $m$, information from the arm pulls is collected, and we estimate the reward of each $C \in \mathcal{A}_m$ by $\widehat{\mu}_m(C) = \frac{1}{n_m} \sum_{i=1}^{n_m} y_{C,i}$. Cubes of low estimated rewards are then eliminated, according to the following rule: a cube $C \in \mathcal{A}_m$ is eliminated if $\widehat{\mu}_m^{\max} - \widehat{\mu}_m(C) \geq 4r_m$, where $\widehat{\mu}_m^{\max} := \max_{C \in \mathcal{A}_m} \widehat{\mu}_m(C)$. After necessary removal of "bad cubes", each cube in $\mathcal{A}_m$ that survives the elimination is equally partitioned into $\left(\frac{r_m}{r_{m+1}}\right)^d$ subcubes of edge

---

[2]One can arbitrarily play $x_{C,1}, x_{C,2}, \cdots, x_{C,n_m}$ as long as $x_{C,i} \in C$ for all $i$.

length $r_{m+1}$, where $r_{m+1}$ is predetermined. These cubes (of edge length $r_{m+1}$) are collected to construct $\mathcal{A}_{m+1}$, and the learning process moves on to the next batch. Appropriate rounding may be required to ensure the ratio $\frac{r_m}{r_{m+1}}$ is an integer. See Remark 2 for more details.

The learning process is summarized in Algorithm 1.

---

**Algorithm 1** Batched Lipschitz Narrowing (BLiN)
---

1: **Input.** Arm set $\mathcal{A} = [0,1]^d$; time horizon $T$.
2: **Initialization** Number of batches $B$; Edge-length sequence $\{r_m\}_{m=1}^{B+1}$; The first grid point $t_0 = 0$; Equally partition $\mathcal{A}$ to $r_1^d$ subcubes and define $\mathcal{A}_1$ as the collection of these subcubes.
3: Compute $n_m = \frac{16 \log T}{r_m^2}$ for $m = 1, \cdots, B+1$.
4: **for** $m = 1, 2, \cdots, B$ **do**
5:     For each cube $C \in \mathcal{A}_m$, play arms $x_{C,1}, \cdots x_{C,n_m}$ from $C$.
6:     Collect the rewards of all pulls up to $t_m$. Compute the average payoff $\widehat{\mu}_m(C) = \frac{\sum_{i=1}^{n_m} y_{C,i}}{n_m}$ for each cube $C \in \mathcal{A}_m$. Find $\widehat{\mu}_m^{max} = \max_{C \in \mathcal{A}_m} \widehat{\mu}(C)$.
7:     For each cube $C \in \mathcal{A}_m$, eliminate $C$ if $\widehat{\mu}_m^{max} - \widehat{\mu}_m(C) > 4r_m$. Let $\mathcal{A}_m^+$ be set of cubes not eliminated.
8:     Compute $t_{m+1} = t_m + (r_m/r_{m+1})^d \cdot |\mathcal{A}_m^+| \cdot n_{m+1}$. If $t_{m+1} \geq T$ or $m = B$ then **break**.
9:     Equally partition each cube in $\mathcal{A}_m^+$ into $(r_m/r_{m+1})^d$ subcubes and define $\mathcal{A}_{m+1}$ as the collection of these subcubes. /\*See Remark 2 for more details on cases where $(r_m/r_{m+1})^d$ is not an integer.\*/
10: **end for**
11: **Cleanup:** Arbitrarily play the remaining arms until all $T$ steps are used.

---

The following theorem gives regret and communication upper bound of BLiN with Doubling Edge-length Sequence $r_m = 2^{-m+1}$ (see Appendix B for proof). Note that this result implies Theorem 1.

**Theorem 3.** *With probability exceeding $1 - \frac{2}{T^6}$, the $T$-step total regret $R(T)$ of BLiN with Doubling Edge-length Sequence (D-BLiN) satisfies*

$$R(T) \leq 528(\log T)^{\frac{1}{d_z+2}} \cdot T^{\frac{d_z+1}{d_z+2}},$$

*where $d_z$ is the zooming dimension of the problem instance. In addition, D-BLiN only needs no more than $\frac{\log T - \log \log T}{d_z+2} + 2$ rounds of communications to achieve this regret rate.*

Although D-BLiN efficiently solves batched Lipschitz bandits, its simple partition strategy leads to suboptimal communication complexity. Now we show that by approporiately combining some batches, BLiN achieves the optimal communication bound, without incurring increasing regret. Specifically, we introduce the following edge-length sequence, which we call ACE Sequence.

**Definition 1.** *For a problem with ambient dimension $d$, zooming dimension $d_z$ and time horizon $T$, we denote $c_1 = \frac{d_z+1}{(d+2)(d_z+2)} \log \frac{T}{\log T}$ and $c_{i+1} = \eta c_i$ for any $i \geq 1$, where $\eta = \frac{d+1-d_z}{d+2}$. Then the Appropriate Combined Edge-length Sequence $\{r_m\}$ is defined by $r_m = 2^{-\sum_{i=1}^{m} c_i}$ for any $m \geq 1$.*

We show that BLiN with ACE Sequence (A-BLiN) obtains an improved communication complexity, thus proves Theorem 2.

**Theorem 4.** *With probability exceeding $1 - \frac{2}{T^6}$, the $T$-step total regret $R(T)$ of Algorithm 1 satisfies*

$$R(T) \leq \left( \frac{128 C_z}{\log \frac{d+2}{d+1-d_z}} \cdot \log \log T + 8e \right) \cdot T^{\frac{d_z+1}{d_z+2}} (\log T)^{\frac{1}{d_z+2}}, \tag{2}$$

*where $d_z$ is the zooming dimension of the problem instance. In addition, Algorithm 1 only needs no more than $\frac{\log \log T}{\log \frac{d+2}{d+1-d_z}} + 1$ rounds of communications to achieve this regret rate.*

The partition and elimination process of a real A-BLiN run is in Figure 1. In the $i$-th subgraph, the white cubes are those remaining after the $(i-1)$-th batch. In this experiment, we set $\mathcal{A} = [0,1]^2$, and the optimal arm is $x^* = (0.8, 0.7)$. Note that $x^*$ is not eliminated during the game. More details of this experiment are in Section 5.

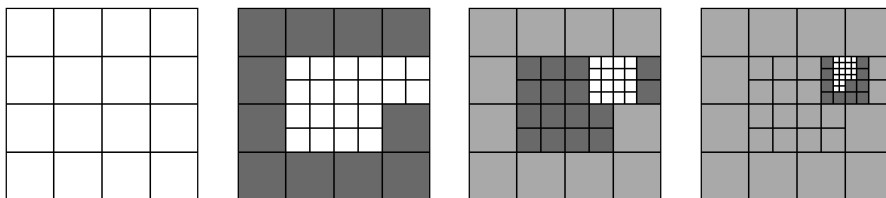

Figure 1: Partition and elimination process of A-BLiN. The $i$-th subfigure shows the pattern before the $i$-th batch, which is from a real A-BLiN run on the reward function defined in Section 5. Dark gray cubes are those eliminated in the most recent batch, while the light gray ones are those eliminated in earlier batches. For the total time horizon $T = 80000$, A-BLiN needs 4 rounds of communications. For this experiment, $r_1 = \frac{1}{4}$, $r_2 = \frac{1}{8}$, $r_3 = \frac{1}{16}$, $r_4 = \frac{1}{32}$, which is the ACE sequence (rounded as in Remark 2) for $d = 2$ and $d_z = 0$.

**Remark 1** (Time and space complexity). *The time complexity of our algorithm is $\mathcal{O}(T)$, which is better than the state of the art $\mathcal{O}(T \log T)$ in [15]. This is because that the running time of a batch $j$ is of order $\mathcal{O}(l_j)$, where $l_j = t_j - t_{j-1}$ is number of samples in batch $j$. Since $\sum_j l_j = T$, the time complexity of BLiN is $\mathcal{O}(T)$. Besides, the space complexity of BLiN is also improved, because we do not need to store information of cubes in previous batches.*

## 3 Regret Analysis of A-BLiN

In this section, we provide regret analysis for A-BLiN. The highlight of the finding is that $\mathcal{O}(\log \log T)$ batches are sufficient to achieve optimal regret rate of $\widetilde{\mathcal{O}}\left(T^{\frac{d_z+1}{d_z+2}}\right)$, as summarized in Theorem 4.

To prove Theorem 4, we first show that the estimator $\widehat{\mu}$ is concentrated to the true expected reward $\mu$ (Lemma 1), and the optimal arm survives all eliminations with high probability (Lemma 2). In the following analysis, we let $B_{\text{stop}}$ be the total number of batches of the BLiN run.

**Lemma 1.** *Define*

$$\mathcal{E} := \left\{ |\mu(x) - \widehat{\mu}_m(C)| \le r_m + \sqrt{\frac{16 \log T}{n_m}}, \forall 1 \le m \le B_{\text{stop}} - 1, \ \forall C \in \mathcal{A}_m, \ \forall x \in C \right\}.$$

*It holds that $\mathbb{P}\left(\mathcal{E}\right) \ge 1 - 2T^{-6}$.*

**Lemma 2.** *Under event $\mathcal{E}$, the optimal arm $x^* = \arg\max \mu(x)$ is not eliminated after the first $B_{\text{stop}} - 1$ batches.*

Based on these results, we show the cubes that survive elimination are of high reward.

**Lemma 3.** *Under event $\mathcal{E}$, for any $1 \le m \le B_{\text{stop}}$, any $C \in \mathcal{A}_m$ and any $x \in C$, $\Delta_x$ satisfies*

$$\Delta_x \le 8r_{m-1}. \tag{3}$$

The proofs of Lemma 1-3 are in Appendix A. We are now ready to prove the theorem.

*Proof of Theorem 4.* Let $R_m$ denote regret of the $m$-th batch. Fixing any positive number $B$, the total regret $R(T)$ can be divided into two parts: $R(T) = \sum_{m \le B} R_m + \sum_{m > B} R_m$. In the following, we bound these two parts separately and then determine $B$ to obtain the upper bound of the total regret. Moreover, we show A-BLiN uses only $\mathcal{O}(\log \log T)$ rounds of communications to achieve the optimal regret.

Recall that $\mathcal{A}_m$ is set of the active cubes in the $m$-th batch. According to Lemma 3, for any $x \in \cup_{C \in \mathcal{A}_m} C$, we have $\Delta_x \le 8r_{m-1}$. Let $\mathcal{A}_m^+$ be set of cubes not eliminated in batch $m$. Then any cube in $\mathcal{A}_{m-1}^+$ is a subset of $S(8r_{m-1})$, and thus

$$|\mathcal{A}_{m-1}^+| \le N_{r_{m-1}} \le C_z r_{m-1}^{-d_z}. \tag{4}$$

By definition, $r_m = r_{m-1}2^{-c_m}$, so

$$|\mathcal{A}_m| = 2^{dc_m}|\mathcal{A}_{m-1}^+|. \tag{5}$$

The total regret of the $m$-th batch is

$$R_m = \sum_{C \in \mathcal{A}_m} \sum_{i=1}^{n_m} \Delta_{x_{C,i}} \le |\mathcal{A}_m| \cdot \frac{16 \log T}{r_m^2} \cdot 8r_{m-1} \overset{(i)}{=} 2^{dc_m}|\mathcal{A}_{m-1}^+| \cdot \frac{16 \log T}{r_m^2} \cdot 8r_{m-1}$$

$$\overset{(ii)}{\le} 2^{dc_m} \cdot C_z r_{m-1}^{-d_z+1} \cdot \frac{128 \log T}{r_m^2} \overset{(iii)}{=} 2^{(\sum_{i=1}^{m-1} c_i)(d_z+1)+c_m(d+2)} \cdot 128 C_z \log T,$$

where (i) follows from (5), (ii) follows from (4), and (iii) follows from the definition of ACE Sequence.

Define $C_m = (\sum_{i=1}^{m-1} c_i)(d_z + 1) + c_m(d + 2)$. Since $c_m = c_{m-1} \cdot \frac{d+1-d_z}{d+2}$, calculation shows that $C_m = (\sum_{i=1}^{m-2} c_i)(d_z + 1) + c_{m-1}(d + 2) + c_{m-1}(d_z + 1 - d - 2) + c_m(d + 2) = C_{m-1}$. Thus for any $m$, we have $C_m = C_1 = c_1(d + 2)$. Hence,

$$R_m \le 2^{c_1(d+2)} \cdot 128 C_z \log T = T^{\frac{d_z+1}{d_z+2}} \cdot 128 C_z (\log T)^{\frac{1}{d_z+2}}. \tag{6}$$

The inequality (6) holds even if the $m$-th batch does not exist (where we let $R_m = 0$) or is not completed. Thus we obtain the first upper bound $\sum_{m \le B} R_m \le T^{\frac{d_z+1}{d_z+2}} \cdot 128 C_z \cdot B(\log T)^{\frac{1}{d_z+2}}$. Lemma 3 implies that any arm $x$ played after the first $B$ batches satisfies $\Delta_x \le 8r_B$, so the total regret after $B$ batches is bounded by

$$\sum_{m > B} R_m \le 8r_B \cdot T = 8T \cdot 2^{-\sum_{i=1}^B c_i} = 8T \cdot 2^{-c_1(\frac{1-\eta^B}{1-\eta})}$$

$$= 8T^{\frac{d_z+1}{d_z+2}}(\log T)^{\frac{1}{d_z+2}} \cdot (T/\log T)^{\frac{\eta^B}{d_z+2}} \le 8T^{\frac{d_z+1}{d_z+2}}(\log T)^{\frac{1}{d_z+2}} \cdot T^{\frac{\eta^B}{d_z+2}}.$$

Therefore, the total regret $R(T)$ satisfies

$$R(T) = \sum_{m \le B} R_m + \sum_{m > B} R_m \le \left(128 C_z \cdot B + 8T^{\frac{\eta^B}{d_z+2}}\right) \cdot T^{\frac{d_z+1}{d_z+2}}(\log T)^{\frac{1}{d_z+2}}.$$

This inequality holds for any positive $B$. Then by choosing $B^* = \frac{\log \log T - \log(d_z+2)}{\log \frac{1}{\eta}} = \frac{\log \log T - \log(d_z+2)}{\log \frac{d+2}{d+1-d_z}}$, we have $\frac{\eta^{B^*}}{d_z+2} = \frac{1}{\log T}$ and

$$R(T) \le \left(\frac{128 C_z \log \log T}{\log \frac{d+2}{d+1-d_z}} + 8e\right) \cdot T^{\frac{d_z+1}{d_z+2}}(\log T)^{\frac{1}{d_z+2}}.$$

The above analysis implies that we can achieve the optimal regret rate $\widetilde{\mathcal{O}}\left(T^{\frac{d_z+1}{d_z+2}}\right)$ by letting the *for-loop* run $B^*$ times and finishing the remaining rounds in the *Cleanup* step. In other words, $B^* + 1$ rounds of communications are sufficient for A-BLiN to achieve the regret bound (2). $\square$

**Remark 2.** *The quantity $\frac{r_m}{r_{m+1}}$ in line 9 of Algorithm 1 may not be integers for some $m$. Thus, in practice we denote $\alpha_n = \lfloor \sum_{i=1}^n c_i \rfloor$, $\beta_n = \lceil \sum_{i=1}^n c_i \rceil$, and define rounded ACE Sequence $\{\widetilde{r}_m\}_{m \in \mathbb{N}}$ by $\widetilde{r}_m = 2^{-\alpha_k}$ for $m = 2k-1$ and $\widetilde{r}_m = 2^{-\beta_k}$ for $m = 2k$. Then the total regret can be divided as $R(T) = \sum_{1 \le k \le B^*} R_{2k-1} + \sum_{1 \le k \le B^*} R_{2k} + \sum_{m > 2B^*} R_m$. For the first part we have $\widetilde{r}_{2k-2} \le r_{k-1}$ and $\widetilde{r}_{2k-1} \ge r_k$, while for the second part we have $\frac{\widetilde{r}_{2k-1}}{\widetilde{r}_{2k}} = 2$. Therefore, by similar argument to the proof of Theorem 4, we can bound these three parts separately, and conclude that BLiN with rounded ACE sequence achieves the optimal regret bound $\widetilde{\mathcal{O}}(T^{\frac{d_z+1}{d_z+2}})$ by using only $\mathcal{O}(\log \log T)$ rounds of communications. The details are in Appendix C.*

# 4 Lower Bounds

In this section, we present lower bounds for Lipschitz bandits with batched feedback, which in turn gives communication lower bounds for all Lipschitz bandit algorithms. Our lower bounds depend on the rounds of communications $B$. When $B$ is sufficiently large, our results match the lower bound for the vanilla Lipschitz bandit problem $\widetilde{\Theta}(R_z(T))$ ($R_z(T)$ is defined in Eq. 1). More importantly, this dependency on $B$ gives the minimal rounds of communications needed to achieve optimal regret bound for all Lipschitz bandit algorithms, which is summarized in Corollary 2. Since this lower bound matches the upper bound presented in Theorem 4, BLiN optimally solves Lipschitz bandits with minimal communication.

## 4.1 Proof Outline

Similar to most lower bound proofs, we need to construct problem instances that are difficult to differentiate. What's different is that we need to carefully integrate batched feedback pattern [33] with the Lipschitz payoff reward [39, 30]. To capture the adaptivity in grid determination, we construct "static reference communication grids" to remove the stochasticity in grid selection [1, 21]. Below, we first consider the static grid case, where the grid is predetermined. This static grid case will provide intuition for the adaptive and more general case.

The expected reward functions of these instances are constructed as follows: we choose some 'position' and 'height', such that the expected reward function obtains local maximum of the specified 'height' at the specified 'position'. We will use the word 'peak' to refer to the local maxima. The following theorem presents the lower bound for the static grid case.

**Theorem 5** (Lower Bound for Static Grid). *Consider Lipschitz bandit problems with time horizon $T$ such that the grid of reward communication $\mathcal{T}$ is static and determined before the game. If $B$ rounds of communications are allowed, then for any policy $\pi$, there exists a problem instance such that*

$$\mathbb{E}[R_T(\pi)] \geq c \cdot (\log T)^{-\frac{\frac{1}{d+2}}{1-\left(\frac{1}{d+2}\right)^B}} \cdot R_z(T)^{\frac{1}{1-\left(\frac{1}{d+2}\right)^B}},$$

*where $c > 0$ is a numerical constant independent of $B$, $T$, $\pi$ and $\mathcal{T}$, $R_z(T)$ is defined in (1), and $d$ is the dimension of the arm space.*

To prove Theorem 5, we first show that for any $k > 1$ there exists an instance such that $\mathbb{E}[R_T(\pi)] \geq \frac{t_k}{t_{k-1}^{\frac{1}{d+2}}}$. Fixing $k > 1$, we let $r_k = \frac{1}{t_{k-1}^{\frac{1}{d+2}}}$ and $M_k := t_{k-1} r_k^2 = \frac{1}{r_k^d}$. Then we construct a set of problem instances $\mathcal{I}_k = \{I_{k,1}, \cdots, I_{k,M_k}\}$, such that the gap between the highest peak and the second highest peak is about $r_k$ for every instance in $\mathcal{I}_k$.

Based on this construction, we prove that no algorithm can distinguish instances in $\mathcal{I}_k$ from one another in the first $(k-1)$ batches, so the worst-case regret is at least $r_k t_k$, which gives the inequality we need. For the first batch $(0, t_1]$, we can easily construct a set of instances where the worst-case regret is at least $t_1$, since no information is available during this time. Thus, there exists a problem instance such that $\mathbb{E}[R_T(\pi)] \gtrsim \max\left\{t_1, \frac{t_2}{t_1^{\frac{1}{d+2}}}, \cdots, \frac{t_B}{t_{B-1}^{\frac{1}{d+2}}}\right\}$. Since $0 < t_1 < \cdots < t_B = T$, the inequality in Theorem 5 follows.

As a result of Theorem 5, we can derive the minimum rounds of communications needed to achieve optimal regret bound for Lipschitz bandit problem, which is stated in Corollary 1.

**Corollary 1.** *Any Lipschitz bandit algorithm needs $\Omega(\log \log T)$ rounds of communications to achieve the optimal regret rate, for the case that the times of reward communication are predetermined and static.*

The detailed proof of Theorem 5 and Corollary 1 are deferred to Appendix D and E.

## 4.2 Communication Lower Bound for BLiN

So far we have derived lower bounds for the static grid case. Yet there is a gap between the static and the adaptive case. We will close this gap in the following Theorem.

**Theorem 6** (Lower Bound for Adaptive Grid). *Consider Lipschitz bandit problems with time horizon $T$ such that the grid of reward communication $\mathcal{T}$ is adaptively determined by the player. If $B$ rounds of communications are allowed, then for any policy $\pi$, there exists an instance such that*

$$\mathbb{E}\left[R_T(\pi)\right] \geq c\frac{1}{B^2}(\log T)^{-\frac{\frac{1}{d+2}}{1-\left(\frac{1}{d+2}\right)^B}} R_z(T)^{\frac{1}{1-\left(\frac{1}{d+2}\right)^B}},$$

*where $c > 0$ is a numerical constant independent of $B$, $T$, $\pi$ and $\mathcal{T}$, $R_z(T)$ is defined in (1), and $d$ is the dimension of the arm space.*

To prove Theorem 6, we consider a reference static grid $\mathcal{T}_r = \{T_0, T_1, \cdots, T_B\}$, where $T_j = T^{\frac{1-\varepsilon^j}{1-\varepsilon^B}}$ for $\varepsilon = \frac{1}{d+2}$. We construct a series of 'worlds', denoted by $\mathcal{I}_1, \cdots, \mathcal{I}_B$. Each world is a set of problem instances, and each problem instance in world $\mathcal{I}_j$ is defined by peak location set $\mathcal{U}_j$ and basic height $r_j$, where the sets $\mathcal{U}_j$ and quantities $r_j$ for $1 \leq j \leq B$ are presented in Appendix F.

Based on these constructions, we first prove that for any adaptive grid and policy, there exists an index $j$ such that the event $A_j = \{t_{j-1} < T_{j-1}, \ t_j \geq T_j\}$ happens with sufficiently high probability in world $\mathcal{I}_j$. Then similar to Theorem 5, we prove that in world $\mathcal{I}_j$ there exists a set of problem instances that is difficult to differentiate in the first $j - 1$ batches. In addition, event $A_j$ implies that $t_j \geq T_j$, so the worst-case regret is at least $r_j T_j$, which gives the lower bound we need.

The proof of Theorem 6 is deferred to Appendix F. Similar to Corollary 1, we can prove that at least $\Omega(\log \log T)$ rounds of communications are needed to achieve optimal regret bound. This result is formally summarized in Corollary 2.

**Corollary 2.** *Any Lipschitz bandit algorithm[3] needs $\Omega(\log \log T)$ rounds of communications to achieve the optimal regret rate.*

## 5 Experiments

In this section, we present numerical studies of A-BLiN. In the experiments, we use the arm space $\mathcal{A} = [0,1]^2$ and the expected reward function $\mu(x) = 1 - \frac{1}{2}\|x - x_1\|_2 - \frac{3}{10}\|x - x_2\|_2$, where $x_1 = (0.8, \ 0.7)$ and $x_2 = (0.1, \ 0.1)$. The landscape of $\mu$ and the resulting partition is shown in Figure 2(a). As can be seen, the partition is finer in the area closer to the optimal arm $x^* = (0.8, \ 0.7)$.

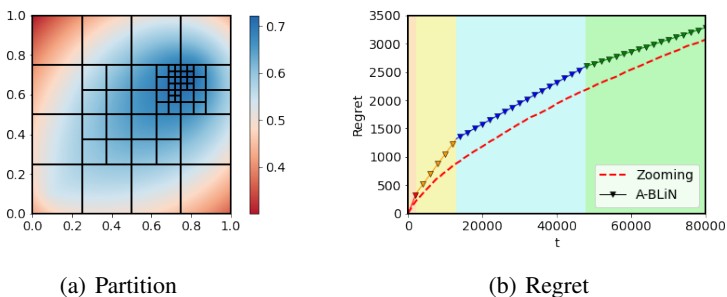

(a) Partition                    (b) Regret

Figure 2: Resulting partition and regret of A-BLiN. In Figure 2(a), we show the resulting partition of A-BLiN. The background color denotes the true value of expected reward $\mu$, and blue means high values. The figure shows that the partition is finer for larger values of $\mu$. In Figure 2(b), we show accumulated regret of A-BLiN and zooming algorithm [26]. In the figure, different background colors represent different batches of A-BLiN. For the total time horizon $T = 80000$, A-BLiN needs 4 rounds of communications.

We let the time horizon $T = 80000$, and report the accumulated regret in Figure 2(b). The regret curve is sublinear, which agrees with the regret bound (2). Besides, different background colors in Figure 2(b) represent different batches. For the total time horizon $T = 80000$, A-BLiN only needs 4 rounds of communications. We also present regret curve of zooming algorithm [26] for

---

[3]including algorithms of which the number of batches can be determined adaptively

comparison. Different from zooming algorithm, regret curve of A-BLiN is approximately piecewise linear, which is because the strategy of BLiN does not change within each batch. Results of more repeated experiments are in Appendix G, as well as experimental results of D-BLiN. Our code is available at `https://github.com/FengYasong-fifol/Batched-Lipschitz-Narrowing`.

## 6 Conclusion

In this paper, we study Lipschitz bandits with communication constraints, and propose the BLiN algorithm as a solution. We prove that BLiN only need $\mathcal{O}\left(\log \log T\right)$ rounds of communications to achieve the optimal regret rate of best previous Lipschitz bandit algorithms [26, 14] that need $T$ batches. This improvement in number of the batches significantly saves data communication costs. We also provide complexity analysis for this problem. We show that $\Omega(\log \log T)$ rounds of communications are necessary for any algorithm to optimally solve Lipschitz bandit problems. Hence, BLiN algorithm is optimal.

## Acknowledgments and Disclosure of Funding

This work was partly supported by the National Key Research and Development Program of China (2020AAA0107600).

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
