*Proof.* Fix a cube $C \in \mathcal{A}_m$. Recall the average payoff of cube $C \in \mathcal{A}_m$ is defined as

$$\widehat{\mu}_m(C) = \frac{\sum_{i=1}^{n_m} y_{C,i}}{n_m}.$$

We also have

$$\mathbb{E}[\widehat{\mu}_m(C)] = \frac{\sum_{i=1}^{n_m} \mu(x_{C,i})}{n_m}.$$

Since $\widehat{\mu}_m(C) - \mathbb{E}[\widehat{\mu}_m(C)]$ obeys normal distribution $\mathcal{N}\left(0, \frac{1}{n_m}\right)$, Hoeffding inequality gives

$$\mathbb{P}\left( |\widehat{\mu}_m(C) - \mathbb{E}[\widehat{\mu}_m(C)]| \ge \sqrt{\frac{16 \log T}{n_m}} \right) \le \frac{2}{T^8}.$$

On the other hand, by Lipschitzness of $\mu$, it is obvious that

$$|\mathbb{E}[\widehat{\mu}_m(C)] - \mu(x)| \le r_m, \quad \forall x \in C.$$

Consequently, we have

$$\mathbb{P}\left( \sup_{x \in C} |\mu(x) - \widehat{\mu}_m(C)| \le r_m + \sqrt{\frac{16 \log T}{n_m}} \right) \ge 1 - \frac{2}{T^8}.$$

For $1 \le m \le B_{\text{stop}} - 1$, the $m$-th batch is finished, so any cube $C \in \mathcal{A}_m$ is played for not less than 1 time, and thus $|\mathcal{A}_m| \le T$. From here, by similar argument to Lemma F.1 in [38] and Lemma 1 in [30], taking a union bound over $C \in \mathcal{A}_m$ and $1 \le m \le B_{\text{stop}} - 1$ finishes the proof. $\square$

**Lemma 2.** Under event $\mathcal{E}$, the optimal arm $x^* = \arg\max \mu(x)$ is not eliminated after the first $B_{\text{stop}} - 1$ batches.

*Proof.* We use $C_m^*$ to denote the cube containing $x^*$ in $\mathcal{A}_m$. Here we proof that $C_m^*$ is not eliminated in round $m$.

Under event $\mathcal{E}$, for any cube $C \in \mathcal{A}_m$ and $x \in C$, we have

$$\widehat{\mu}(C) - \widehat{\mu}(C_m^*) \le \mu(x) + \sqrt{\frac{16 \log T}{n_m}} + r_m - \mu(x^*) + \sqrt{\frac{16 \log T}{n_m}} + r_m \le 4r_m.$$

Then from the elimination rule, $C_m^*$ is not eliminated. $\square$

**Lemma 3.** Under event $\mathcal{E}$, for any $1 \le m \le B_{\text{stop}}$, any $C \in \mathcal{A}_m$ and any $x \in C$, $\Delta_x$ satisfies

$$\Delta_x \le 8r_{m-1}.$$

*Proof.* For $m = 1$, (3) holds directly from the Lipschitzness of $\mu$. For $m > 1$, let $C_{m-1}^*$ be the cube in $\mathcal{A}_{m-1}$ such that $x^* \in C_{m-1}^*$. From Lemma 2, this cube $C_{m-1}^*$ is well-defined under $\mathcal{E}$. For any cube $C \in \mathcal{A}_m$ and $x \in C$, it is obvious that $x$ is also in the parent of $C$ (the cube in the previous round that contains $C$), which is denoted by $C_{par}$. Thus for any $x \in C$, it holds that

$$\Delta_x = \mu^* - \mu(x) \le \widehat{\mu}_{m-1}(C_{m-1}^*) + \sqrt{\frac{16 \log T}{n_{m-1}}} + r_{m-1} - \widehat{\mu}_{m-1}(C_{par}) + \sqrt{\frac{16 \log T}{n_{m-1}}} + r_{m-1},$$

where the inequality uses Lemma 1.

Equality $\sqrt{\frac{16 \log T}{n_{m-1}}} = r_{m-1}$ gives that

$$\Delta_x \leq \widehat{\mu}_{m-1}(C_{m-1}^*) - \widehat{\mu}_{m-1}(C_{par}) + 4r_{m-1}.$$

It is obvious that $\widehat{\mu}_{m-1}(C_{m-1}^*) \leq \widehat{\mu}_{m-1}^{\max}$. Moreover, since the cube $C_{par}$ is not eliminated, from the elimination rule we have

$$\widehat{\mu}_{m-1}^{\max} - \widehat{\mu}_{m-1}(C_{par}) \leq 4r_{m-1}.$$

Hence, we conclude that $\Delta_x \leq 8r_{m-1}$. $\qquad\square$

## B  Proof of Theorem 3

**Theorem 3.** With probability exceeding $1 - \frac{2}{T^6}$, the $T$-step total regret $R(T)$ of BLiN with Doubling Edge-length Sequence (D-BLiN) satisfies

$$R(T) \leq 528(\log T)^{\frac{1}{d_z+2}} \cdot T^{\frac{d_z+1}{d_z+2}}, \tag{7}$$

where $d_z$ is the zooming dimension of the problem instance. In addition, D-BLiN only needs no more than $\frac{\log T - \log \log T}{d_z+2} + 2$ rounds of communications to achieve this regret rate.

*Proof.* Since $r_m = \frac{r_{m-1}}{2}$ for Doubling Edge-length Sequence, Lemma 3 implies that every cube $C \in \mathcal{A}_m$ is a subset of $S(16r_m)$. Thus from the definition of zooming number, we have

$$|\mathcal{A}_m| \leq N_{r_m}. \tag{8}$$

Fix any positive number $B$. Also by Lemma 3, we know that any arm played after batch $B$ incurs a regret bounded by $16r_B$, since the cubes played after batch $B$ have edge length no larger than $r_B$. Then the total regret occurs after the first $B$ batch is bounded by $16r_B T$.

Thus the regret $R(T)$ can be bounded by

$$R(T) \leq \sum_{m=1}^{B} \sum_{C \in \mathcal{A}_m} \sum_{i=1}^{n_m} \Delta_{x_{C,i}} + 16r_B T, \tag{9}$$

where the first term bounds the regret in the first $B$ batches of D-BLiN, and the second term bounds the regret after the first $B$ batches. If the algorithm stops at batch $\widetilde{B} < B$, we define $\mathcal{A}_m = \varnothing$ for any $\widetilde{B} < m \leq B$ and inequality (9) still holds.

By Lemma 3, we have $\Delta_{C,i} \leq 16r_m$ for all $C \in \mathcal{A}_m$. We can thus bound (9) by

$$R(T) \leq \sum_{m=1}^{B} |\mathcal{A}_m| \cdot n_m \cdot 16r_m + 16r_B T$$

$$\leq \sum_{m=1}^{B} N_{r_m} \cdot n_m \cdot 16r_m + 16r_B T, \tag{10}$$

$$\leq \sum_{m=1}^{B} N_{r_m} \cdot \frac{16 \log T}{r_m^2} \cdot 16r_m + 16r_B T, \tag{11}$$

$$= \sum_{m=1}^{B} N_{r_m} \cdot \frac{256 \log T}{r_m} + 16r_B T,$$

where (10) uses (8), and (11) uses equality $n_m = \frac{16 \log T}{r_m}$. Since $r_m = 2^{-m+1}$ and $N_{r_m} \leq r_m^{-d_z} \leq 2^{(m-1)d_z}$, we have

$$R(T) \leq 256 \sum_{m=1}^{B} \frac{2^{(m-1)d_z} \log T}{2^{-(m-1)}} + 16 \cdot 2^{-B+1} T.$$

This inequality holds for any positive $B$. By choosing $B^* = 1 + \frac{\log \frac{T}{\log T}}{d_z + 2}$, we have

$$
\begin{aligned}
R(T) \leq & 512 \cdot 2^{(B^*-1)(d_z+1)} \log T + 16 \cdot T \cdot 2^{-B^*+1} \\
\leq & 528 T^{\frac{d_z+1}{d_z+2}} (\log T)^{\frac{1}{d_z+2}}.
\end{aligned}
$$

The above analysis implies that we can achieve the optimal regret rate $\widetilde{\mathcal{O}}\left(T^{\frac{d_z+1}{d_z+2}}\right)$ by letting the *for-loop* run $B^*$ times and finishing the remaining rounds in the *Cleanup* step. In other words, $B^* + 1$ rounds of communications are sufficient for D-BLiN to achieve the regret bound (7). $\qquad\square$

## C  Proof of Remark 2

For rounded ACE Sequence, the quantity $\frac{\widetilde{r}_m}{\widetilde{r}_{m+1}}$ is an integer for any $m$, so the partition in Line 9 of Algorithm 1 is well-defined. In this section, we show that BLiN with rounded ACE sequence can also achieve the optimal regret bound by using $\mathcal{O}(\log \log T)$ batches. For any $m$, if there exists $k < m$ such that $\widetilde{r}_m \geq \widetilde{r}_k$, then we skip the $m$-th batch. It is easy to verify that the following analysis is still valid in this case.

**Theorem 7.** *With probability exceeding $1 - \frac{2}{T^6}$, the $T$-step total regret $R(T)$ of Algorithm 1 with rounded ACE sequence satisfies*

$$
R(T) \leq \left( \frac{128 C_z \log \log T}{\log \frac{d+2}{d+1-d_z}} + 512 C_z + 8e \right) \cdot T^{\frac{d_z+1}{d_z+2}} (\log T)^{\frac{1}{d_z+2}},
$$

*where $d_z$ is the zooming dimension of the problem instance. In addition, Algorithm 1 only needs no more than $\frac{2 \log \log T}{\log \frac{d+2}{d+1-d_z}} + 1$ rounds of communications to achieve this regret rate.*

*Proof.* The proof of Theorem 7 is similar to that of Theorem 4.

Firstly, we fix positive number $B^* = \frac{\log \log T - \log(d_z+2)}{\log \frac{d+2}{d+1-d_z}}$ and consider the first $2B^*$ batches. As is summairzed in Remark 2, we bound the regret caused by the first $2B^*$ batches through two different arguments.

For $m = 2k - 1$, $1 \leq k \leq B^*$, we have $\widetilde{r}_m = 2^{-\alpha_k}$ and $\widetilde{r}_{m-1} = 2^{-\beta_{k-1}}$, and thus

$$
\widetilde{r}_m \geq r_k \quad \text{and} \quad \widetilde{r}_{m-1} \leq r_{k-1}. \tag{12}
$$

Let $\mathcal{A}_m^+$ be set of cubes not eliminated in round $m$. Similar argument to Theorem 4 shows that $|\mathcal{A}_{m-1}^+| \leq C_z \widetilde{r}_{m-1}^{-d_z}$. The total regret of round $m$ is

$$
\begin{aligned}
R_m &= \sum_{C \in \mathcal{A}_m} \sum_{i=1}^{n_m} \Delta_{x_{C,i}} \\
&\leq |\mathcal{A}_m| \cdot \frac{16 \log T}{\widetilde{r}_m^2} \cdot 8 \widetilde{r}_{m-1} \\
&= \left( \frac{\widetilde{r}_{m-1}}{\widetilde{r}_m} \right)^d |\mathcal{A}_{m-1}^+| \cdot \frac{16 \log T}{\widetilde{r}_m^2} \cdot 8 \widetilde{r}_{m-1} \\
&\leq \left( \frac{\widetilde{r}_{m-1}}{\widetilde{r}_m} \right)^d \cdot C_z \widetilde{r}_{m-1}^{-d_z} \cdot \frac{16 \log T}{\widetilde{r}_m^2} \cdot 8 \widetilde{r}_{m-1} \\
&\leq \widetilde{r}_{m-1}^{d+1-d_z} \cdot \widetilde{r}_m^{-d-2} \cdot 128 C_z \log T \\
&\leq r_{k-1}^{d+1-d_z} \cdot r_k^{-d-2} \cdot 128 C_z \log T \\
&= T^{\frac{d_z+1}{d_z+2}} \cdot 128 C_z \cdot (\log T)^{\frac{1}{d_z+2}},
\end{aligned}
$$

where the sixth line follows from (12), and the seventh line follows from (6). Summing over $k$ gives that

$$
\sum_{k=1}^{B^*} R_{2k-1} \leq T^{\frac{d_z+1}{d_z+2}} \cdot 128 C_z \cdot B^* \cdot (\log T)^{\frac{1}{d_z+2}} \leq \frac{128 C_z \log \log T}{\log \frac{d+2}{d+1-d_z}} \cdot T^{\frac{d_z+1}{d_z+2}} (\log T)^{\frac{1}{d_z+2}}. \tag{13}
$$

For $m = 2k$, $1 \le k \le B^*$, we have $\widetilde{r}_m = 2^{-\beta_k}$ and $\widetilde{r}_{m-1} = 2^{-\alpha_k}$, and thus $\widetilde{r}_m = \frac{1}{2}\widetilde{r}_{m-1}$. Lemma 3 shows that any cube in $\mathcal{A}_m$ is a subset of $S(16\widetilde{r}_m)$, so we have $|\mathcal{A}_m| \le N_{\widetilde{r}_m} \le C_z \widetilde{r}_m^{-d_z}$. Therefore, the total regret of round $m$ is

$$
\begin{aligned}
R_m &= \sum_{C \in \mathcal{A}_m} \sum_{i=1}^{n_m} \Delta_{x,C_i} \\
&\le |\mathcal{A}_m| \cdot \frac{16 \log T}{\widetilde{r}_m^2} \cdot 16\widetilde{r}_m \\
&\le C_z \widetilde{r}_m^{-d_z} \cdot \frac{16 \log T}{\widetilde{r}_m^2} \cdot 16\widetilde{r}_m \\
&= \widetilde{r}_m^{-d_z-1} \cdot 256 C_z \log T.
\end{aligned}
$$

Since $\frac{\widetilde{r}_{2k-2}}{\widetilde{r}_{2k}} = \frac{\widetilde{r}_{2k-2}}{\widetilde{r}_{2k-1}} \cdot \frac{\widetilde{r}_{2k-1}}{\widetilde{r}_{2k}} \ge 2$, summing over $k$ gives that

$$
\sum_{k=1}^{B^*} R_{2k} \le \widetilde{r}_{2B^*}^{-d_z-1} \cdot 512 C_z \log T.
$$

The definition of round ACE Sequence shows that

$$
\widetilde{r}_{2B^*} = 2^{-\lceil \sum_{i=1}^{B^*} c_i \rceil} = 2^{-\left\lceil c_1 \left( \frac{1 - \eta^{B^*}}{1 - \eta} \right) \right\rceil} = 2^{-\left\lceil \frac{\log \frac{T}{\log T}}{d_z + 2} - 1 \right\rceil} \ge \left( \frac{T}{\log T} \right)^{-\frac{1}{d_z + 2}},
$$

so we have

$$
\sum_{k=1}^{B^*} R_{2k} \le \left( \left( \frac{T}{\log T} \right)^{-\frac{1}{d_z+2}} \right)^{-d_z - 1} \cdot 512 C_z \log T = T^{\frac{d_z+1}{d_z+2}} \cdot 512 C_z (\log T)^{\frac{1}{d_z+2}}. \tag{14}
$$

Similar argument to Theorem 4 shows that the total regret after $2B^*$ batches is upper bounded by $8\widetilde{r}_{2B^*} T$. Since $\widetilde{r}_{2B^*} \le r_{B^*}$, we further have

$$
\sum_{m \ge 2B^*} R_m \le 8\widetilde{r}_{2B^*} T \le 8 r_{B^*} T \le 8e \cdot T^{\frac{d_z+1}{d_z+2}} (\log T)^{\frac{1}{d_z+2}}. \tag{15}
$$

Combining (13), (14) and (15), we conclude that

$$
R(T) \le \left( \frac{128 C_z \log \log T}{\log \frac{d+2}{d+1-d_z}} + 512 C_z + 8e \right) \cdot T^{\frac{d_z+1}{d_z+2}} (\log T)^{\frac{1}{d_z+2}}.
$$

The analysis in Theorem 7 implies that we can achieve the optimal regret rate $\widetilde{\mathcal{O}}\left( T^{\frac{d_z+1}{d_z+2}} \right)$ by letting the *for-loop* of Algorithm 1 run $2B^*$ times and finishing the remaining rounds in the *Cleanup* step. In other words, $2B^* + 1$ rounds of communications are sufficient for BLiN to achieve the optimal regret. $\qquad\square$

## D  Proof of Theorem 5

**Theorem 5.** Consider Lipschitz bandit problems with time horizon $T$ such that the grid of reward communication $\mathcal{T}$ is static and determined before the game. If $B$ rounds of communications are allowed, then for any policy $\pi$, there exists a problem instance such that

$$
\mathbb{E}[R_T(\pi)] \ge c \cdot (\log T)^{-\frac{\frac{1}{d+2}}{1 - \left( \frac{1}{d+2} \right)^B}} \cdot R_z(T)^{\frac{1}{1 - \left( \frac{1}{d+2} \right)^B}},
$$

where $c > 0$ is a numerical constant independent of $B$, $T$, $\pi$ and $\mathcal{T}$, $R_z(T)$ is defined in (1), and $d$ is the dimension of the arm space.

*Proof.* To prove Theorem 5, we first show that for any $k > 1$, there exists an instance such that $\mathbb{E}[R_T(\pi)] \geq \frac{t_k}{t_{k-1}^{\frac{1}{d+2}}}$.

Fixing an index $k > 1$, we construct a set of problem instances that is difficult to distinguish. Let $r_k = \frac{1}{t_{k-1}^{\frac{1}{d+2}}}$ and $M_k := t_{k-1}r_k^2 = \frac{1}{r_k^d}$. We can find a set of arms $\mathcal{U}_k = \{u_{k,1}, \cdots, u_{k,M_k}\}$ such that $d_{\mathcal{A}}(u_{k,i}, u_{k,j}) \geq r_k$ for any $i \neq j$. Then we consider a set of problem instances $\mathcal{I}_k = \{I_{k,1}, \cdots, I_{k,M_k}\}$. The expected reward for $I_{k,1}$ is defined as

$$
\mu_{k,1}(x) = \begin{cases} \dfrac{3}{4}r_k, & x = u_{k,1}, \\[2mm] \dfrac{5}{8}r_k, & x = u_{k,j}, \ j \neq 1, \\[2mm] \max\left\{\dfrac{r_k}{2}, \max_{u \in \mathcal{U}_k}\{\mu_{k,1}(u) - d_{\mathcal{A}}(x,u)\}\right\}, & otherwise. \end{cases} \tag{16}
$$

For $2 \leq i \leq M_k$, the expected reward for $I_{k,i}$ is defined as

$$
\mu_{k,i}(x) = \begin{cases} \dfrac{3}{4}r_k, & x = u_{k,1}, \\[2mm] \dfrac{7}{8}r_k, & x = u_{k,i}, \\[2mm] \dfrac{5}{8}r_k, & x = u_{k,j}, \ j \neq 1 \ and \ j \neq i, \\[2mm] \max\left\{\dfrac{r_k}{2}, \max_{u \in \mathcal{U}_k}\{\mu_{k,i}(u) - d_{\mathcal{A}}(x,u)\}\right\}, & otherwise. \end{cases} \tag{17}
$$

For all arm pulls in all problem instances, an Gaussian noise sampled from $\mathcal{N}(0,1)$ is added to the observed reward. This noise corruption is independent from all other randomness.

The lower bound of expected regret relies on the following lemma.

**Lemma 4.** *For any policy $\pi$, there exists a problem instance $I \in \mathcal{I}_k$ such that*

$$
\mathbb{E}\left[R_T(\pi)\right] \geq \frac{r_k}{32} \cdot \sum_{j=1}^{B}(t_j - t_{j-1})\exp\left\{-\frac{t_{j-1}r_k^2}{32(M_k - 1)}\right\}.
$$

*Proof.* Let $S_{k,i} = \mathbb{B}(u_{k,i}, \frac{3}{8}r_k)$ (the ball with center $u_{k,i}$ and radius $\frac{3}{8}r_k$). It is easy to verify the following properties of construction (16) and (17):

1. For any $2 \leq i \leq M_k$, $\mu_{k,i}(x) = \mu_{k,1}(x)$ for any $x \in \mathcal{A} - S_{k,i}$;

2. For any $2 \leq i \leq M_k$, $\mu_{k,1}(x) \leq \mu_{k,i}(x) \leq \mu_{k,1}(x) + \frac{r_k}{4}$, for any $x \in S_{k,i}$;

3. For any $1 \leq i \leq M_k$, under $I_{k,i}$, pulling an arm that is not in $S_{k,i}$ incurs a regret at least $\frac{r_k}{8}$.

Let $x_t$ denote the choices of policy $\pi$ at time $t$, and $y_t$ denote the reward. Additionally, for $t_{j-1} < t \leq t_j$, we define $\mathbb{P}_{k,i}^t$ as the distribution of sequence $(x_1, y_1, \cdots, x_{t_{j-1}}, y_{t_{j-1}})$ under instance $I_{k,i}$ and policy $\pi$. It holds that

$$
\begin{aligned} \sup_{I \in \mathcal{I}_k} \mathbb{E}R_T(\pi) &\geq \frac{1}{M_k}\sum_{i=1}^{M_k}\mathbb{E}_{\mathbb{P}_{k,i}}\left[R_T(\pi)\right] \\ &\geq \frac{1}{M_k}\sum_{i=1}^{M_k}\sum_{t=1}^{T}\mathbb{E}_{\mathbb{P}_{k,i}^t}\left[R^t(\pi)\right] \\ &\geq \frac{r_k}{8}\sum_{t=1}^{T}\frac{1}{M_k}\sum_{i=1}^{M_k}\mathbb{P}_{k,i}^t(x_t \notin S_{k,i}), \end{aligned} \tag{18}
$$

where $R^t(\pi)$ denotes the regret incurred by policy $\pi$ at time $t$.

From our construction, it is easy to see that $S_{k,j} \cap S_{k,j} = \varnothing$, so we can construct a test $\Psi$ such that $x_t \in S_{k,i}$ implies $\Psi = i$. Then from Lemma 8,

$$\frac{1}{M_k} \sum_{i=1}^{M_k} \mathbb{P}_{k,i}^t(x_t \notin S_{k,i}) \geq \frac{1}{M_k} \sum_{i=1}^{M_k} \mathbb{P}_{k,i}^t(\Psi \neq i) \geq \frac{1}{2M_k} \sum_{i=2}^{M_k} \exp\left\{-D_{KL}\left(\mathbb{P}_{k,1}^t \| \mathbb{P}_{k,i}^t\right)\right\}.$$

Now we calculate $D_{KL}\left(\mathbb{P}_{k,1}^t \| \mathbb{P}_{k,i}^t\right)$. From the chain rule of KL-Divergence, we have

$$
\begin{aligned}
D_{KL}\left(\mathbb{P}_{k,1}^t \| \mathbb{P}_{k,i}^t\right) =& D_{KL}\left(\mathbb{P}_{k,1}^t(x_1, y_1, \cdots, x_{t_{j-1}}, y_{t_{j-1}}) \| \mathbb{P}_{k,i}^t(x_1, y_1, \cdots, x_{t_{j-1}}, y_{t_{j-1}})\right) \\
=& D_{KL}\left(\mathbb{P}_{k,1}^t(x_1, y_1, \cdots, x_{t_{j-1}-1}, y_{t_{j-1}-1}, x_{t_{j-1}}) \| \mathbb{P}_{k,i}^t(x_1, y_1, \cdots, x_{t_{j-1}-1}, y_{t_{j-1}-1}, x_{t_{j-1}})\right) \\
& + \mathbb{E}_{\mathbb{P}_{k,1}}\left(D_{KL}\left(\mathbb{P}_{k,1}^t(y_{t_{j-1}}|x_1, y_1, \cdots, x_{t_{j-1}}) \| \mathbb{P}_{k,i}^t(y_{t_{j-1}}|x_1, y_1, \cdots, x_{t_{j-1}})\right)\right) \\
\leq& D_{KL}\left(\mathbb{P}_{k,1}^t(x_1, y_1, \cdots, x_{t_{j-1}-1}, y_{t_{j-1}-1}) \| \mathbb{P}_{k,i}^t(x_1, y_1, \cdots, x_{t_{j-1}-1}, y_{t_{j-1}-1})\right) \\
& + \mathbb{E}_{\mathbb{P}_{k,1}}\left(D_{KL}\left(\mathbb{P}_{k,1}^t(y_{t_{j-1}}|x_{t_{j-1}}) \| \mathbb{P}_{k,i}^t(y_{t_{j-1}}|x_{t_{j-1}})\right)\right), \quad (19) \\
\leq& D_{KL}\left(\mathbb{P}_{k,1}^t(x_1, y_1, \cdots, x_{t_{j-1}-1}, y_{t_{j-1}-1}) \| \mathbb{P}_{k,i}^t(x_1, y_1, \cdots, x_{t_{j-1}-1}, y_{t_{j-1}-1})\right) \\
& + \mathbb{E}_{\mathbb{P}_{k,1}}\left(D_{KL}\left(N(\mu_{k,1}(x_{t_{j-1}}), 1) \| N(\mu_{k,i}(x_{t_{j-1}}), 1)\right)\right), \quad (20) \\
=& D_{KL}\left(\mathbb{P}_{k,1}^t(x_1, y_1, \cdots, x_{t_{j-1}-1}, y_{t_{j-1}-1}) \| \mathbb{P}_{k,i}^t(x_1, y_1, \cdots, x_{t_{j-1}-1}, y_{t_{j-1}-1})\right) \\
& + \mathbb{E}_{\mathbb{P}_{k,1}}\left(\frac{1}{2}\left(\mu_{k,1}(x_{t_{j-1}}) - \mu_{k,i}(x_{t_{j-1}})\right)^2\right), \\
\leq& D_{KL}\left(\mathbb{P}_{k,1}^t(x_1, y_1, \cdots, x_{t_{j-1}-1}, y_{t_{j-1}-1}) \| \mathbb{P}_{k,i}^t(x_1, y_1, \cdots, x_{t_{j-1}-1}, y_{t_{j-1}-1})\right) \\
& + \mathbb{E}_{\mathbb{P}_{k,1}}\left(\mathbf{1}_{\{x_{t_{j-1}} \in S_{k,i}\}} \cdot \frac{1}{2}\left(\frac{r_k}{4}\right)^2\right) \quad (21) \\
=& D_{KL}\left(\mathbb{P}_{k,1}^t(x_1, y_1, \cdots, x_{t_{j-1}-1}, y_{t_{j-1}-1}) \| \mathbb{P}_{k,i}^t(x_1, y_1, \cdots, x_{t_{j-1}-1}, y_{t_{j-1}-1})\right) \\
& + \frac{r_k^2}{32} \cdot \mathbb{P}_{k,1}\left(x_{t_{j-1}} \in S_{k,i}\right), \quad (22)
\end{aligned}
$$

where (19) uses the non-negativity of KL-Divergence and the conditional independence of the reward, (20) uses that the rewards are corrupted by a standard normal noise, and (21) uses the first two properties of the construction.

From (22), we then decompose the KL-Divergence step by step and conclude that

$$D_{KL}\left(\mathbb{P}_{k,1}^t \| \mathbb{P}_{k,i}^t\right) \leq \frac{r_k^2}{32} \cdot \sum_{s \leq t_{j-1}} \mathbb{P}_{k,1}\left(x_s \in S_{k,i}\right) = \frac{r_k^2}{32} \mathbb{E}_{\mathbb{P}_{k,1}} \tau_i, \quad (23)$$

where $\tau_i$ denotes the number of pulls of arms in $S_{k,i}$ before the batch containing $t$. Then for all $t \in (t_{j-1}, t_j]$, we have

$$
\begin{aligned}
\frac{1}{M_k} \sum_{i=1}^{M_k} \mathbb{P}_{k,i}^t(x_t \notin S_{k,i}) &\geq \frac{1}{2M_k} \sum_{i=2}^{M_k} \exp\left\{-\frac{r_k^2}{32} \mathbb{E}_{\mathbb{P}_{k,1}} \tau_i\right\} \\
&\geq \frac{M_k - 1}{2M_k} \exp\left\{-\frac{r_k^2}{32(M_k - 1)} \sum_{i=2}^{M_k} \mathbb{E}_{\mathbb{P}_{k,1}} \tau_i\right\} \quad (24) \\
&\geq \frac{1}{4} \exp\left\{-\frac{r_k^2 t_{j-1}}{32(M_k - 1)}\right\}, \quad (25)
\end{aligned}
$$

where (24) uses the Jensen' inequality, and (25) uses the fact that $\sum_{i=2}^{M_k} \tau_i \leq t_{j-1}$. Finally, we substitute (25) to (18) to finish the proof. $\qquad \square$

Since $M_k = t_{k-1} r_k^2$, the expected regret of policy $\pi$ satisfies

$$\mathbb{E}\left[R_T(\pi)\right] \geq \frac{r_k}{32} \cdot \sum_{j=1}^{B} (t_j - t_{j-1}) \exp\left\{ -\frac{t_{j-1} r_k^2}{32(M_k - 1)} \right\}$$

$$\geq \frac{r_k}{32} \cdot \sum_{j=1}^{B} (t_j - t_{j-1}) \exp\left\{ -\frac{t_{j-1} r_k^2}{16 M_k} \right\}$$

$$\geq \frac{r_k}{32} \cdot \sum_{j=1}^{B} (t_j - t_{j-1}) \exp\left\{ -\frac{t_{j-1}}{16 t_{k-1}} \right\}$$

on instance $I$ defnied in Lemma 4.

By omitting terms with $j > k$ in the above summation, we have

$$\mathbb{E}[R_T(\pi)] \geq \frac{r_k}{32} \cdot \sum_{j=1}^{B} (t_j - t_{j-1}) \exp\left\{ -\frac{t_{j-1}}{16 t_{k-1}} \right\}$$

$$\geq \frac{r_k}{32} \cdot \sum_{j=1}^{k} (t_j - t_{j-1}) \exp\left\{ -\frac{1}{16} \right\}$$

$$= \frac{1}{32 e^{\frac{1}{16}}} r_k t_k$$

$$= \frac{1}{32 e^{\frac{1}{16}}} \cdot \frac{t_k}{t_{k-1}^{\frac{1}{d+2}}}.$$

The above analysis can be applied for any $k > 1$. For the first batch $(0, t_1]$, we can easily construct a set of instances where the worst-case regret is at least $t_1$, since no information is available during this time. Thus, there exists a problem instance such that

$$\mathbb{E}[R_T(\pi)] \geq \frac{1}{32 e^{\frac{1}{16}}} \max\left\{ t_1, \frac{t_2}{t_1^{\frac{1}{d+2}}}, \cdots, \frac{t_B}{t_{B-1}^{\frac{1}{d+2}}} \right\}.$$

Since $0 < t_1 < \cdots < t_B = T$, we further have

$$\mathbb{E}\left[R_T(\pi)\right] \geq \frac{1}{32 e^{\frac{1}{16}}} \cdot T^{\frac{1 - \frac{1}{d+2}}{1 - \left(\frac{1}{d+2}\right)^B}}. \tag{26}$$

Since $N_r \leq r^{-d}$ holds for all instances,

$$\sum_{r = 2^{-i}, \, r \geq r_0} \frac{N_r}{r} \log T \leq 2 r_0^{-d-1} \log T.$$

Then we have

$$R_z(T) = \inf_{r_0} \left\{ r_0 T + \sum_{r = 2^{-i}, \, r \geq r_0} \frac{N_r}{r} \log T \right\}$$

$$\leq \inf_{r_0} \left\{ r_0 T + \frac{2}{r_0^{d+1}} \log T \right\} \tag{27}$$

$$\leq 2 (\log T)^{\frac{1}{d+2}} T^{1 - \frac{1}{d+2}}.$$

As a consequence, for instance $I$ satisfying (26),

$$\mathbb{E}[R_T(\pi)] \geq \frac{1}{32e^{\frac{1}{16}}} \left( \frac{1}{2(\log T)^{\frac{1}{d+2}}} \right)^{\frac{1}{1-\left(\frac{1}{d+2}\right)^B}} \cdot R_z(T)^{\frac{1}{1-\left(\frac{1}{d+2}\right)^B}}$$

$$\geq \frac{1}{32e^{\frac{1}{16}} \cdot 2^{\frac{1}{1-\frac{1}{d+2}}}} \cdot (\log T)^{-\frac{\frac{1}{d+2}}{1-\left(\frac{1}{d+2}\right)^B}} \cdot R_z(T)^{\frac{1}{1-\left(\frac{1}{d+2}\right)^B}}$$

$$\geq \frac{1}{128e^{\frac{1}{16}}} \cdot (\log T)^{-\frac{\frac{1}{d+2}}{1-\left(\frac{1}{d+2}\right)^B}} \cdot R_z(T)^{\frac{1}{1-\left(\frac{1}{d+2}\right)^B}}.$$

Hence, the proof is completed. $\qquad\square$

# E   Proof of Corollary 1

**Corollary 1.**   Any algorithm needs $\Omega(\log\log T)$ rounds of communications to optimally solve Lipschitz bandit problems, for the case that the times of reward communication are predetermined and static.

*Proof.* From (26), the expected regret is lower bounded by $\frac{1}{32e^{\frac{1}{16}}} \cdot T^{\frac{1-\frac{1}{d+2}}{1-\left(\frac{1}{d+2}\right)^B}}$. When $B$ is sufficiently large, the bound becomes $\frac{1}{32e^{\frac{1}{16}}} \cdot T^{1-\frac{1}{d+2}}$. Here we seek for the minimum $B$ such that

$$\frac{\frac{1}{32e^{\frac{1}{16}}} \cdot T^{\frac{1-\frac{1}{d+2}}{1-\left(\frac{1}{d+2}\right)^B}}}{\frac{1}{32e^{\frac{1}{16}}} \cdot T^{1-\frac{1}{d+2}}} \leq C \tag{28}$$

for some constant $C$.

Calculation shows that

$$\frac{\frac{1}{32e^{\frac{1}{16}}} \cdot T^{\frac{1-\frac{1}{d+2}}{1-\left(\frac{1}{d+2}\right)^B}}}{\frac{1}{32e^{\frac{1}{16}}} \cdot T^{1-\frac{1}{d+2}}} = \left( T^{\frac{d+1}{d+2}} \right)^{\frac{1}{(d+2)^B - 1}}. \tag{29}$$

Substituting (29) to (28) and taking log on both sides yield that

$$\frac{d+1}{d+2} \cdot \frac{\log T}{(d+2)^B - 1} \leq \log C$$

and

$$(d+2)^B \geq \frac{d+1}{(d+2)\log C} \cdot \log T + 1.$$

Taking log on both sides again yields that

$$B \geq \frac{\log\left[ \left( \frac{d+1}{(d+2)\log C} \right) \log T + 1 \right]}{\log(d+2)}.$$

Therefore, $\Omega(\log\log T)$ rounds of communications are necessary for any algorithm to optimally solve Lipschitz bandit problems. $\qquad\square$

# F   Proof of Theorem 6

**Theorem 6.** Consider Lipschitz bandit problems with time horizon $T$ such that the grid of reward communication $\mathcal{T}$ is adaptively determined by the player. If $B$ rounds of communications are allowed, then for any policy $\pi$, there exists an instance such that

$$\mathbb{E}\left[R_T(\pi)\right] \geq c\frac{1}{B^2}(\log T)^{-\frac{\frac{1}{d+2}}{1-\left(\frac{1}{d+2}\right)^B}} R_z(T)^{\frac{1}{1-\left(\frac{1}{d+2}\right)^B}},$$

where $c > 0$ is a numerical constant independent of $B$, $T$, $\pi$ and $\mathcal{T}$, $R_z(T)$ is defined in (1), and $d$ is the dimension of the arm space.

*Proof.* Firstly, we define $r_j = \frac{1}{T_{j-1}^{\varepsilon}B}$ and $M_j = \frac{1}{r_j^d}$. From the definition, we have

$$T_{j-1}r_j^2 = \frac{1}{r_j^d B^2} = \frac{M_j}{B^2}. \tag{30}$$

For $1 \leq j \leq B$, we can find sets of arms $\mathcal{U}_j = \{u_{j,1}, \cdots, u_{j,M_j}\}$ such that (a) $d_{\mathcal{A}}(u_{j,m}, u_{j,n}) \geq r_j$ for any $m \neq n$, and (b) $u_{1,M_1} = \cdots = u_{B,M_B}$.

Then we present the construction of worlds $\mathcal{I}_1, \cdots, \mathcal{I}_B$. For $1 \leq j \leq B-1$, we let $\mathcal{I}_j = \{I_{j,k}\}_{k=1}^{M_j-1}$, and the expected reward of $I_{j,k}$ is defined as

$$\mu_{j,k}(x) = \begin{cases} \dfrac{r_1}{2} + \dfrac{r_j}{16} + \dfrac{r_B}{16}, & x = u_{j,k}, \\[2mm] \dfrac{r_1}{2} + \dfrac{r_B}{16}, & x = u_{j,M_j}, \end{cases} \tag{31}$$

and $\mu_{j,k}(x) = \max\left\{\frac{r_1}{2}, \max_{u \in \mathcal{U}_j}\{\mu_{j,k}(u) - d_{\mathcal{A}}(x,u)\}\right\}$, otherwise. For $j = B$, we let $\mathcal{I}_B = \{I_B\}$. The expected reward of $I_B$ is defined as

$$\mu_B(u_{B,M_B}) = \frac{r_1}{2} + \frac{r_B}{16} \tag{32}$$

and $\mu_B(x) = \max\left\{\frac{r_1}{2}, \mu_B(u_{B,M_B}) - d_{\mathcal{A}}(x, u_{B,M_B})\right\}$, otherwise.

As mentioned above, based on these constructions, we first show that for any adaptive grid $\mathcal{T} = \{t_0, \cdots, t_B\}$, there exists an index $j$ such that $(t_{j-1}, t_j]$ is sufficiently large in world $\mathcal{I}_j$. More formally, for each $j \in [B]$, and event $A_j = \{t_{j-1} < T_{j-1}, \ t_j \geq T_j\}$, we define the quantities $p_j := \frac{1}{M_j-1}\sum_{k=1}^{M_j-1}\mathbb{P}_{j,k}(A_j)$ for $j \leq B-1$ and $p_B := \mathbb{P}_B(A_B)$, where $\mathbb{P}_{j,k}(A_j)$ denotes the probability of the event $A_j$ under instance $I_{j,k}$ and policy $\pi$. For these quantities, we have the following lemma.

**Lemma 5.** *For any adaptive grid $\mathcal{T}$ and policy $\pi$, it holds that $\sum_{j=1}^{B} p_j \geq \frac{7}{8}$.*

*Proof.* For $1 \leq j \leq B-1$ and $1 \leq k \leq M_j-1$, we define $S_{j,k} = \mathbb{B}(u_{j,k}, \frac{3}{8}r_j)$, which is the ball centered as $u_{j,k}$ with radius $\frac{3}{8}r_j$. It is easy to verify the following properties of our construction (31) and (32):

1. $\mu_{j,k}(x) = \mu_B(x)$ for any $x \notin S_{j,k}$;

2. $\mu_B(x) \leq \mu_{j,k}(x) \leq \mu_B(x) + \frac{r_j}{8}$, for any $x \in S_{j,k}$.

Let $x_t$ denote the choices of policy $\pi$ at time $t$, and $y_t$ denote the reward. For $t_{j-1} < t \leq t_j$, we define $\mathbb{P}_{j,k}^t$ (resp. $\mathbb{P}_B^t$) as the distribution of sequence $(x_1, y_1, \cdots, x_{t_{j-1}}, y_{t_{j-1}})$ under instance $I_{j,k}$ (resp. $I_B$) and policy $\pi$. Since event $A_j$ can be completely described by the observations up to time $T_{j-1}$ ($A_j$ is an event in the $\sigma$-algebra where $\mathbb{P}_{j,k}^{T_{j-1}}$ and $\mathbb{P}_B^{T_{j-1}}$ are defined on), we can use the definition of total variation to get

$$|\mathbb{P}_B(A_j) - \mathbb{P}_{j,k}(A_j)| = |\mathbb{P}_B^{T_{j-1}}(A_j) - \mathbb{P}_{j,k}^{T_{j-1}}(A_j)| \leq TV\left(\mathbb{P}_B^{T_{j-1}}, \mathbb{P}_{j,k}^{T_{j-1}}\right).$$

For the total variation, we apply Lemma 7 to get

$$\frac{1}{M_j - 1} \sum_{k=1}^{M_j-1} TV\left(\mathbb{P}_B^{T_{j-1}}, \mathbb{P}_{j,k}^{T_{j-1}}\right) \le \frac{1}{M_j - 1} \sum_{k=1}^{M_j-1} \sqrt{1 - \exp\left(-D_{KL}\left(\mathbb{P}_B^{T_{j-1}} \| \mathbb{P}_{j,k}^{T_{j-1}}\right)\right)}.$$

An argument similar to (23) yields that

$$D_{KL}\left(\mathbb{P}_B^{T_{j-1}} \| \mathbb{P}_{j,k}^{T_{j-1}}\right) \le \frac{r_j^2}{128} \mathbb{E}_{\mathbb{P}_B} \tau_k,$$

where $\tau_k$ denotes the number of pulls which is in $S_{j,k}$ before the batch containing $T_{j-1}$. Combining the above two inequalities gives

$$\frac{1}{M_j - 1} \sum_{k=1}^{M_j-1} TV\left(\mathbb{P}_B^{T_{j-1}}, \mathbb{P}_{j,k}^{T_{j-1}}\right) \le \frac{1}{M_j - 1} \sum_{k=1}^{M_j-1} \sqrt{1 - \exp\left(-\frac{r_j^2}{128}\mathbb{E}_{\mathbb{P}_B}\tau_k\right)}$$

$$\le \sqrt{1 - \exp\left(-\frac{r_j^2}{128(M_j-1)}\mathbb{E}_{\mathbb{P}_B}\left[\sum_{k=1}^{M_j-1} \tau_k\right]\right)} \quad (33)$$

$$\le \sqrt{1 - \exp\left(-\frac{r_j^2 T_{j-1}}{128(M_j-1)}\right)} \quad (34)$$

$$\le \sqrt{1 - \exp\left(-\frac{1}{64B^2}\right)} \quad (35)$$

$$\le \frac{1}{8B},$$

where (33) uses Jensen's inequality, (34) uses the fact that $\sum_{k=1}^{M_j-1} \tau_k \le T_{j-1}$, and (35) uses (30).

Plugging the above results implies that

$$|\mathbb{P}_B(A_j) - p_j| \le \frac{1}{M_j - 1} \sum_{k=1}^{M_j-1} |\mathbb{P}_B(A_j) - \mathbb{P}_{j,k}(A_j)| \le \frac{1}{8B}.$$

Since $\sum_{j=1}^{B} \mathbb{P}(A_j) \ge \mathbb{P}\left(\cup_{j=1}^{B} A_j\right) = 1$, it holds that

$$\sum_{j=1}^{B} p_j \ge \mathbb{P}_B(A_M) + \sum_{j=1}^{B-1}\left(\mathbb{P}_B(A_j) - \frac{1}{8B}\right) \ge \frac{7}{8}. \qquad \square$$

Lemma 5 implies that there exists some $j$ such that $p_j > \frac{7}{8B}$. Then similar to Theorem 5, we show that the worst-case regret of the policy in world $\mathcal{I}_j$ gives the lower bound we need.

**Lemma 6.** *For adaptive grid $\mathcal{T}$ and policy $\pi$, if index $j$ satisfies $p_j \ge \frac{7}{8B}$, then there exists a problem instance $I$ such that $\mathbb{E}\left[R_T(\pi)\right] \ge c\frac{1}{B^2}(\log T)^{-\frac{\frac{1}{d+2}}{1-\left(\frac{1}{d+2}\right)^B}} R_z(T)^{\frac{1}{1-\left(\frac{1}{d+2}\right)^B}}$, where $c > 0$ is a numerical constant independent of $B$, $T$, $\pi$ and $\mathcal{T}$.*

*Proof.* Here we proceed with the case where $j \le B - 1$. The case for $j = B$ can be proved analogously.

For any $1 \le k \le M_j - 1$, we construct a set of problem instances $\mathcal{I}_{j,k} = (I_{j,k,l})_{1 \le l \le M_j}$. For $l \ne k$, the expected reward of $I_{j,k,l}$ is defined as

$$\mu_{j,k,l}(x) = \begin{cases} \mu_{j,k}(x) + \dfrac{3r_j}{16}, & x = u_{j,l}, \\[2mm] \mu_{j,k}(x), & x \in \mathcal{U}_j \ \text{ and } \ x \ne u_{j,l}, \\[2mm] \max\left\{\dfrac{r_1}{2}, \max\limits_{u \in \mathcal{U}_j}\{\mu_{j,k,l}(u) - d_{\mathcal{A}}(x, u)\}\right\}, & otherwise, \end{cases}$$

where $\mu_{j,k}$ is defined in (31). For $l = k$, we let $\mu_{j,k,k} = \mu_{j,k}$.

We define $C_{j,k} = \mathbb{B}\left(u_{j,k}, \frac{r_j}{4}\right)$, and our construction $\mathcal{I}_{j,k}$ has the following properties:

1. For any $l \neq k$, $\mu_{j,k,l}(x) = \mu_{j,k,k}(x)$ for any $x \notin C_{j,l}$;

2. For any $l \neq k$, $\mu_{j,k,k}(x) \leq \mu_{j,k,l}(x) \leq \mu_{j,k,k}(x) + \frac{3r_j}{16}$ for any $x \in C_{j,l}$;

3. For any $1 \leq l \leq M_j$, under $I_{j,k,l}$, pulling an arm that is not in $C_{j,l}$ incurs a regret at least $\frac{r_j}{16}$.

Let $x_t$ denote the choices of policy $\pi$ at time $t$, and $y_t$ denote the reward. For $t_{j-1} < t \leq t_j$, we define $\mathbb{P}_{j,k,l}^t$ as the distribution of sequence $(x_1, y_1, \cdots, x_{t_{j-1}}, y_{t_{j-1}})$ under instance $I_{i,j,k}$ and policy $\pi$. From similar argument in (18), it holds that

$$\sup_{I \in \mathcal{I}_{j,k}} \mathbb{E}\left[R_T(\pi)\right] \geq \frac{r_j}{16} \sum_{t=1}^{T} \frac{1}{M_j} \sum_{l=1}^{M_j} \mathbb{P}_{j,k,l}^t(x_t \notin C_{j,l}). \tag{36}$$

From our construction, it is easy to see that $C_{j,k_1} \cap C_{j,k_2} = \varnothing$ for any $k_1 \neq k_2$, so we can construct a test $\Psi$ such that $x_t \notin C_{j,k}$ implies $\Psi \neq k$. By Lemma 8 with a star graph on $[K]$ with center $k$, we have

$$\frac{1}{M_j} \sum_{l=1}^{M_j} \mathbb{P}_{j,k,l}^t(x_t \notin C_{j,l}) \geq \frac{1}{M_j} \sum_{l \neq k} \int \min\left\{d\mathbb{P}_{j,k,k}^t, d\mathbb{P}_{j,k,l}^t\right\}. \tag{37}$$

Combining (36) and (37) gives

$$\sup_{I \in \mathcal{I}_{j,k}} \mathbb{E}\left[R_T(\pi)\right] \geq \frac{r_j}{16} \sum_{t=1}^{T} \frac{1}{M_j} \sum_{l \neq k} \int \min\left\{d\mathbb{P}_{j,k,k}^t, d\mathbb{P}_{j,k,l}^t\right\}$$

$$\geq \frac{r_j}{16} \sum_{t=1}^{T_j} \frac{1}{M_j} \sum_{l \neq k} \int \min\left\{d\mathbb{P}_{j,k,k}^t, d\mathbb{P}_{j,k,l}^t\right\}$$

$$\geq \frac{r_j T_j}{16} \cdot \frac{1}{M_j} \sum_{l \neq k} \int \min\left\{d\mathbb{P}_{j,k,k}^{T_j}, d\mathbb{P}_{j,k,l}^{T_j}\right\} \tag{38}$$

$$\geq \frac{r_j T_j}{16} \cdot \frac{1}{M_j} \sum_{l \neq k} \int_{A_j} \min\left\{d\mathbb{P}_{j,k,k}^{T_j}, d\mathbb{P}_{j,k,l}^{T_j}\right\} \tag{39}$$

$$\geq \frac{r_j T_j}{16} \cdot \frac{1}{M_j} \sum_{l \neq k} \int_{A_j} \min\left\{d\mathbb{P}_{j,k,k}^{T_{j-1}}, d\mathbb{P}_{j,k,l}^{T_{j-1}}\right\}, \tag{40}$$

where (38) follows from data processing inequality of total variation and the equation $\int \min\{dP, dQ\} = 1 - TV(P, Q)$, (39) restricts the integration to event $A_j$, and (40) holds because the observations at time $T_j$ are the same as those at time $T_{j-1}$ under event $A_j$.

For the term $\int_{A_j} \min\left\{d\mathbb{P}_{j,k,k}^{T_{j-1}}, d\mathbb{P}_{j,k,l}^{T_{j-1}}\right\}$, it holds that

$$\int_{A_j} \min\left\{d\mathbb{P}_{j,k,k}^{T_{j-1}}, d\mathbb{P}_{j,k,l}^{T_{j-1}}\right\} = \int_{A_j} \frac{d\mathbb{P}_{j,k,k}^{T_{j-1}} + d\mathbb{P}_{j,k,l}^{T_{j-1}} - \left|d\mathbb{P}_{j,k,k}^{T_{j-1}} - d\mathbb{P}_{j,k,l}^{T_{j-1}}\right|}{2}$$

$$= \frac{\mathbb{P}_{j,k,k}^{T_{j-1}}(A_j) + \mathbb{P}_{j,k,l}^{T_{j-1}}(A_j)}{2} - \frac{1}{2} \int_{A_j} \left|d\mathbb{P}_{j,k,k}^{T_{j-1}} - d\mathbb{P}_{j,k,l}^{T_{j-1}}\right|$$

$$\geq \left(\mathbb{P}_{j,k,k}^{T_{j-1}}(A_j) - \frac{1}{2} TV\left(\mathbb{P}_{j,k,k}^{T_{j-1}}, \mathbb{P}_{j,k,l}^{T_{j-1}}\right)\right) - TV\left(\mathbb{P}_{j,k,k}^{T_{j-1}}, \mathbb{P}_{j,k,l}^{T_{j-1}}\right) \tag{41}$$

$$= \mathbb{P}_{j,k}(A_j) - \frac{3}{2} TV\left(\mathbb{P}_{j,k,k}^{T_{j-1}}, \mathbb{P}_{j,k,l}^{T_{j-1}}\right), \tag{42}$$

where (41) uses the inequality $|\mathbb{P}(A) - \mathbb{Q}(A)| \le TV(\mathbb{P}, \mathbb{Q})$, and (42) holds because $I_{j,k} = I_{j,k,k}$ and $A_j$ can be determined by the observations up to $T_{j-1}$.

We use an argument similar to (23) to get

$$
D_{KL}\left(\mathbb{P}_{j,k,k}^{T_{j-1}} \| \mathbb{P}_{j,k,l}^{T_{j-1}}\right) \le \frac{1}{2} \cdot \left(\frac{3r_j}{16}\right)^2 \mathbb{E}_{\mathbb{P}_{j,k}} \tau_l \le \frac{r_j^2}{32} \mathbb{E}_{\mathbb{P}_{j,k}} \tau_l,
$$

where $\tau_l$ denotes the number of pulls which is in $S_{j,l}$ before the batch of time $T_{j-1}$. Then from Lemma 7, we have

$$
\begin{aligned}
\frac{1}{M_j} \sum_{l \ne k} TV\left(\mathbb{P}_{j,k,k}^{T_{j-1}}, \mathbb{P}_{j,k,l}^{T_{j-1}}\right) &\le \frac{1}{M_j} \sum_{l \ne k} \sqrt{1 - \exp\left(-D_{KL}\left(\mathbb{P}_{j,k,k}^{T_{j-1}} \| \mathbb{P}_{j,k,l}^{T_{j-1}}\right)\right)} \\
&\le \frac{1}{M_j} \sum_{l \ne k} \sqrt{1 - \exp\left(-\frac{r_j^2}{32} \mathbb{E}_{\mathbb{P}_{j,k}} \tau_l\right)} \\
&\le \frac{M_j - 1}{M_j} \sqrt{1 - \exp\left(-\frac{r_j^2}{32(M_j-1)} \sum_{l \ne k} \mathbb{E}_{\mathbb{P}_{j,k}} \tau_l\right)} \\
&\le \frac{M_j - 1}{M_j} \sqrt{1 - \exp\left(-\frac{r_j^2 T_{j-1}}{32(M_j-1)}\right)} \\
&\le \frac{M_j - 1}{M_j} \sqrt{1 - \exp\left(-\frac{M_j}{32(M_j-1)B^2}\right)} \qquad (43) \\
&\le \frac{M_j - 1}{M_j} \sqrt{\frac{M_j}{32(M_j-1)B^2}} \\
&\le \frac{1}{4B}, \qquad\qquad\qquad\qquad\qquad\qquad\qquad (44)
\end{aligned}
$$

where (43) uses (30).

Combining (40), (42) and (44) yields that

$$
\begin{aligned}
\sup_{I \in \mathcal{I}_{j,k}} \mathbb{E}\left[R_T(\pi)\right] &\ge \frac{1}{16} r_j T_j \left(\frac{\mathbb{P}_{j,k}(A_j)}{2} - \frac{3}{8B}\right) \\
&\ge \frac{1}{16B} T^{\frac{1-\varepsilon}{1-\varepsilon^B}} \left(\frac{\mathbb{P}_{j,k}(A_j)}{2} - \frac{3}{8B}\right).
\end{aligned}
$$

This inequality holds for any $k \le M_j - 1$. Averaging over $k$ yields

$$
\begin{aligned}
\sup_{I \in \cup_{k \le M_j - 1} \mathcal{I}_{j,k}} \mathbb{E}\left[R_T(\pi)\right] &\ge \frac{1}{16B} T^{\frac{1-\varepsilon}{1-\varepsilon^B}} \left(\frac{1}{2(M_j-1)} \sum_{k=1}^{M_j-1} \mathbb{P}_{j,k}(A_j) - \frac{3}{8B}\right) \\
&\ge \frac{1}{16B} T^{\frac{1-\varepsilon}{1-\varepsilon^B}} \left(\frac{7}{16B} - \frac{3}{8B}\right) \qquad (45) \\
&\ge \frac{1}{256B^2} T^{\frac{1-\varepsilon}{1-\varepsilon^B}},
\end{aligned}
$$

where the second inequality holds from $p_j \ge \frac{7}{8B}$.

Consequently, combining (45) and (27) yields that

$$\sup_{I \in \cup_{k \leq M_j - 1} \mathcal{I}_{j,k}} \mathbb{E}\left[R_T(\pi)\right] \geq \frac{1}{256 B^2} T^{\frac{1 - \frac{1}{d+2}}{1 - \left(\frac{1}{d+2}\right)^B}}$$

$$\geq \frac{1}{256 \cdot 2^{\frac{1}{1 - \frac{1}{d+2}}}} \frac{1}{B^2} (\log T)^{-\frac{\frac{1}{d+2}}{1 - \left(\frac{1}{d+2}\right)^B}} R_z(T)^{\frac{1}{1 - \left(\frac{1}{d+2}\right)^B}}$$

$$\geq \frac{1}{1024} \cdot \frac{1}{B^2} (\log T)^{-\frac{\frac{1}{d+2}}{1 - \left(\frac{1}{d+2}\right)^B}} R_z(T)^{\frac{1}{1 - \left(\frac{1}{d+2}\right)^B}},$$

which finishes the proof. □

Finally, combining the above two lemmas, we arrive at the lower bound in Theorem 6. □

## G   Experimental results

### G.1   Repeated experiments of A-BLiN

We present results of A-BLiN with some random seeds below (the curve of zooming algorithm in Figure 2(b) in the paper is the average of 10 repeated experiments), where the figure legends and labels are the same as whose in Figure 2(b). These results stably agree with the plot in the paper. The reason we did not present averaged regret curve of A-BLiN in Figure 2(b) is that we want to show the batch pattern of a single A-BLiN run in the figure. Averaging across different runs breaks the batch pattern. As an example, one stochastic run may end the first batch after 100 observations, while another may end the third batch after 110 observations.

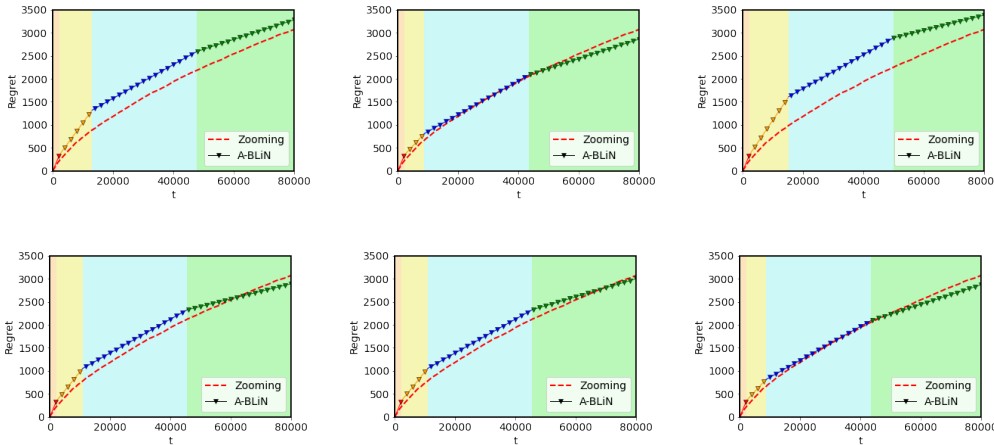

Figure 3: Results of A-BLiN with some random seeds. The figure legends and labels are the same as whose in Figure 2(b).

### G.2   Experimental results of D-BLiN

We run D-BLiN to solve the same problem in Section 5. The partition and elimination process of this experiment is presented in Figure 4, which shows that the optimal arm $x^*$ is not eliminated during the game, and only 6 rounds of communications are needed for time horizon $T = 80000$. Moreover, we present the resulting partition and the accumulated regret in Figure 5.

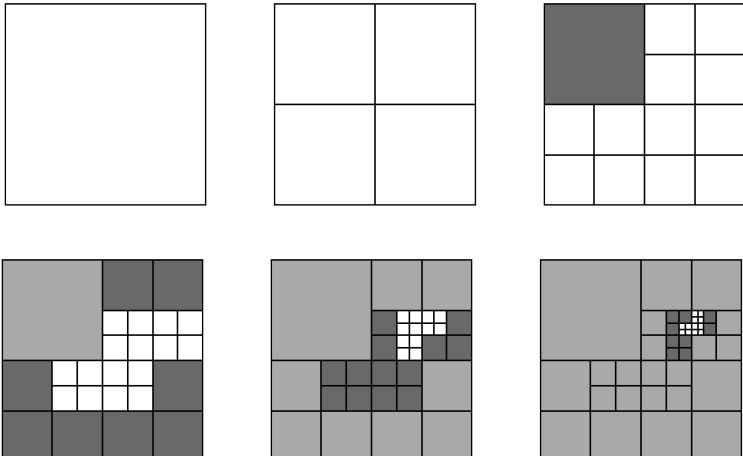

Figure 4: Partition and elimination process of D-BLiN. The $i$-th subfigure shows the pattern before the $i$-th batch. Dark gray cubes are those eliminated in the most recent batch, while the light gray ones are those eliminated in earlier batches. For the total time horizon $T = 80000$, D-BLiN needs 6 rounds of communications.

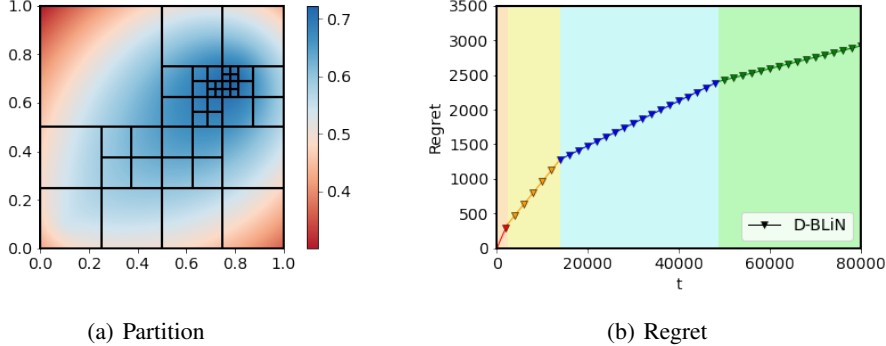

(a) Partition

(b) Regret

Figure 5: Resulting partition and regret of D-BLiN. In Figure 5(a), we show the resulting partition of D-BLiN. The background color denotes the true value of expected reward $\mu$, and blue means high values. The figure shows that the partition is finer for larger values of $\mu$. In Figure 5(b), we show accumulated regret of D-BLiN. In the figure, different background colors represent different batches. For the total time horizon $T = 80000$, D-BLiN needs 6 rounds of communications (the first two batches are too small and are combined with the third batch in the plot).

## H  Auxiliary Technical Tools

**Lemma 7** (Bretagnolle-Huber Inequality[12])**.** *Let $P$ and $Q$ be any probability measures on the same probability space. It holds that*

$$TV(P,Q) \leq \sqrt{1 - \exp\left(-D_{KL}(P\|Q)\right)} \leq 1 - \frac{1}{2}\exp\left(-D_{KL}(P\|Q)\right).$$

**Lemma 8** ([21])**.** *Let $Q_1, \cdots, Q_n$ be probability measures over a common probability space $(\Omega, \mathcal{F})$, and $\Psi : \Omega \to [n]$ be any measurable function (i.e., test). Then for any tree $T = ([n], E)$ with vertex set $[n]$ and edge set $E$, we have*

*1.* $\frac{1}{n}\sum_{i=1}^{n} Q_i(\Psi \neq i) \geq \frac{1}{n}\sum_{(i,j)\in E} \int \min\{dQ_i, dQ_j\}$;

*2.* $\frac{1}{n}\sum_{i=1}^{n} Q_i(\Psi \neq i) \geq \frac{1}{2n}\sum_{(i,j)\in E} \exp\left(-D_{KL}(Q_i\|Q_j)\right)$.