# OpenReview forum: "Lipschitz Bandits with Batched Feedback"
_NeurIPS.cc/2022/Conference — NeurIPS 2022 Accept_

### Official Review · Reviewer_XwZm · 2022-07-06

**Rating:** 6
**Confidence:** 3
**Soundness:** 4 excellent
**Presentation:** 3 good
**Contribution:** 3 good

**Summary:**

This paper provides a batched algorithm for Lipschitz bandit optimization. In each batch, the algorithm divides the remaining domain into cubes, makes a certain amount of arbitrary pulls in each cube, and eliminates sub-optimal cubes based on the noisy observations. This method achieves optimal cumulative regret of $\widetilde{O}(T^\frac{d_z+1}{d_z+2})$ in terms of zooming dimension $d_z$ using $O(\log\log T)$ batches. This work also provides lower bounds for general algorithms of B batches with predetermined and adaptively determined batch allocations, showing $\Omega(\log\log T)$ batches is necessary for algorithms with optimal regret.

Generally, the paper is well-written. The proposed method is easy to understand and has good theoretical analysis.

**Questions:**

- What are the advantages of the proposed algorithms over other Lipschitz bandit algorithms with optimal regret except for the number of batches?
- The algorithm arbitrarily pulls $n_m$ arms in each cube. Does uniformly selecting arms with equal distance make more sense?
- Executing A-BLiN requires knowing the zooming dimension to compute the ACE sequence. Can we obtain the value of the zooming dimension when the true function $\mu$ is unknown?
- In figure 2(b), the slope (or simple regret) of the last batch is not zero. Is there any evidence showing the remaining cubes after the last batch have small sub-optimalities?
- Based on my understanding, the ACE sequence can be predetermined without randomness as in definition 1. Why the sequences of different trials are different in figure 3?

**Limitations:**

There are some discussions regarding the time and space complexity. In the experiments, only one existing algorithm is used for comparison and even has better regret than the proposed algorithm. I recommend trying some different T and therefore different B.

**Strengths And Weaknesses:**

Strengths:
- The algorithm is evaluated comprehensively based on its performance, time complexity, and space complexity.
- The appendix shows significant efforts in theoretical analysis.
- The results are also supported by experiments.

Weakness:
- We need to know the zooming dimension in advance if we wish to use the ACE sequence to achieve $O(\log\log T)$ batches.
- I think the constant term in theorems 3 and 4 might be too large for a 1-Lipschitz function on $[0,1]^d$.
- The practical performance is slightly worse than the baseline algorithm in figure 2(b).

Minor comments:
- I recommend rephrasing theorems 1 and 2 into the normal text since this is the introductory section and theorems 3 and 4 will show the details.

Some works worth mentioning:
- A Domain-Shrinking based Bayesian Optimization Algorithm with Order-Optimal Regret Performance
- Gaussian Process Bandit Optimization with Few Batches

---

> ### Author Response · Authors · 2022-08-02
> **Author Response**
>
> We would like to thank the reviewer for the effort reviewing our paper and the suggestions for improvement. We explain the reviewer's concerns point by point.
> >The practical performance is slightly worse than the baseline algorithm in figure 2(b).
>
> Figure 2(b) is drawn from only one BLiN run. In our repeated experiments, the performance of BLiN can be better than Zooming algorithm (see Figure 3 in Appendix G). Theoretically, since Zooming algorithm has achieved the optimal regret rate, we do not expect our new algorithm to perform better in terms of regret. In this paper, our goal is to achieve competitive regret by using significantly fewer batches.
>
> >I recommend rephrasing theorems 1 and 2 into the normal text since this is the introductory section and theorems 3 and 4 will show the details.
>
> Thanks for the writing suggestion. We will revise our paper accordingly in the final version. We did not rephrase them in the current revision, since deleting the first two theorems will change the theorem indices, which may lead to misunderstandings in our discussion.
>
> >Some works worth mentioning:
> >* A Domain-Shrinking based Bayesian Optimization Algorithm with Order-Optimal Regret Performance
> >* Gaussian Process Bandit Optimization with Few Batches
>
> Thanks for the reference pointers. We have cited these works in the revised version.
>
> >What are the advantages of the proposed algorithms over other Lipschitz bandit algorithms with optimal regret except for the number of batches?
>
> Time complexity of BLiN $\mathcal{O}(T)$ is better than other Lipschitz algorithms, and is theoretically optimal (since time complexity of sampling for any algorithm is $\Omega(T)$). We believe that better time complexity is helpful for practical applications of our algorithm.
>
> >The algorithm arbitrarily pulls $n_m$ arms in each cube. Does uniformly selecting arms with equal distance make more sense?
>
> By uniformly selecting arms, the algorithm may be more stable in practice. However, there is no theoretical advantage to this operation under our problem setting.
>
> >Executing A-BLiN requires knowing the zooming dimension to compute the ACE sequence. Can we obtain the value of the zooming dimension when the true function $\mu$ is unknown?
>
> It is not clear that if the optimal communication bound can be achieved without knowledge of $d_z$. We think this is a good question, and will consider it in future works.
>
> >In figure 2(b), the slope (or simple regret) of the last batch is not zero. Is there any evidence showing the remaining cubes after the last batch have small sub-optimalities?
>
> The slope of the last batch is about $0.021$. The slope does not seem to be very small, probably because the scales of $x$ and $y$ axes in Figure 2(b) are different: $x$-axis ranges from $0$ to $80,000$, while $y$-axis ranges from $0$ to $3,500$. Besides, the background color of Figure 2(a) gives evidence that the remaining cubes have small sub-optimalities.
>
> >Based on my understanding, the ACE sequence can be predetermined without randomness as in definition 1. Why the sequences of different trials are different in figure 3?
>
> The ACE Sequences $\\{r_m\\}$ are predetermined and the same in different trials. What differs between different trials is the time grid $\\{t_m\\}$. This is because that the number of remaining cubes $|A_m^+|$ after each batch $m$ is stochastic, and the next grid point can be written as $t_{m+1}=t_m+(r_m/r_{m+1})^d\cdot|A_m^+|\cdot n_{m+1}$.
>
> >There are some discussions regarding the time and space complexity. In the experiments, only one existing algorithm is used for comparison and even has better regret than the proposed algorithm. I recommend trying some different $T$ and therefore different $B$.
>
> The current work mainly focuses on theoretical aspects of the problem, and the experiments are provided to confirm our theoretical results. We will include more comprehensive experimental results in future works.

---

> > ### Comment · Reviewer_XwZm · 2022-08-07
> > **Acknowledgement of Rebuttal**
> >
> > Thanks for your response and I have no further questions. I would recommend mentioning the challenge of obtaining the zooming dimension as a limitation of A-BLiN.

---

> > > ### Author Response · Authors · 2022-08-09
> > > **Author Response**
> > >
> > > Thank you for your recommendation. We will add it in our final version.

---

### Official Review · Reviewer_Bnts · 2022-07-10

**Rating:** 7
**Confidence:** 4
**Soundness:** 3 good
**Presentation:** 3 good
**Contribution:** 3 good

**Summary:**

This paper studies Lipschitz bandits with batched feedback where the expected reward is a Lipschitz function of arm and the feedback is accessible only in the end of each batch. The paper proposes elimination-based algorithms and derives the regret bound of the proposed algorithms. Lower bounds are also provided to show that the proposed algorithm is nearly optimal in terms of regret and number of batches. Experiments on synthetic dataset demonstrate that the proposed algorithm can achieve regret comparable to existing Lipschitz bandits algorithm with immediate feedback.

**Questions:**

Can the number of cubes over T rounds be upper bounded? What is the time and space complexity of cube partition over T rounds?

**Limitations:**

The potential negative societal impact of this work is ignorable.

**Strengths And Weaknesses:**

Strengths:
This paper is the first to study the batched setting in the context of Lipschitz bandits. The proposed algorithms are novel in the sense that most of existing Lipschitz bandits algorithms are based on the zooming method while this paper takes the elimination method as the algorithmic framework. The proposed algorithms are nealy optimal as both upper bounds and matching lower bounds are established. The empirical performance on synthetic dataset is also promising.

Weakness:
While the paper discuss the time complexity, it ignores the time cost of cube partition. The motivation of the batched setting originates from real-world data but the proposed algorithms are only tested on synthetic datat.

---

> ### Author Response · Authors · 2022-08-02
> **Author Response**
>
> We would like to thank the reviewer for the effort reviewing our paper, the clear summarization and the recognition of our novelty. The question is responded below.
> >Can the number of cubes over $T$ rounds be upper bounded? What is the time and space complexity of cube partition over $T$ rounds?
>
> Thank you for the question. We will show that time and space complexity of cube partition can also be upper bounded by $T$. We let $B$ be the total number of batches of a BLiN run, and $t_m$ be the total number of samples after the $m$-th batch. Firstly, the number of cubes generated before the $(B-1)$-th batch is upper bounded by $T$, since each of these cubes is played for not less than $1$ time during the first $(B-1)$ batches. Secondly, we show that the partition after the $(B-1)$-th batch can be omitted by adding an if-condition. After the elimination of each batch $m$, and before the partition, we can calculate the next time grid by $t_{m+1}=t_m+(r_m/r_{m+1})^d\cdot|A_m^+|\cdot n_{m+1}$. We note that the $(m+1)$-th batch is the last if and only if $t_{m+1}\geq T$. Therefore, we add the following if-condition `If $t_{m+1}\geq T$, then arbitrarily play the remaining arms until all $T$ steps are used', and the partition before the last batch is omitted.
>
> In summary, by adding this if-condition in Algorithm 1, the time and space complexity of cube partition can be upper bounded by $T$. We have modified our algorithm as above in the revised version.

---

### Official Review · Reviewer_ADGD · 2022-07-11

**Rating:** 6
**Confidence:** 4
**Soundness:** 4 excellent
**Presentation:** 3 good
**Contribution:** 2 fair

**Summary:**

The paper considers the problem of Lipschitz-bandits when the evaluations arrive in batches. Two algorithms are proposed. The more adaptive one achieves log-optimal regret with an optimal number of batches. The theoretical part provides regret upper bounds for the algorithms, and lower bounds for static and adaptive grid algorithms. Small empirical evaluations provide illustration on the partition mechanism if the algorithms and a comparison with the zooming algorithm.


**Questions:**

How would be the bounds affected with a different Lipschitz constant, or if the constant would be unknown?

While there is some indication in e.g. line 85 of what the authors mean by optimality, it would be useful to specify it more formally, since it is referred in some theoretical statements (e.g. Corollary 2)


**Limitations:**

The lower bound for the adaptive case relies on the choice of number of batches (B). This would exclude algorithms that adapts the number of batches depending on feedback. This is not a major limitation, and the proofs might be adapted to include this more adaptive variant, but the limitation could be mentioned by the authors.


**Strengths And Weaknesses:**

The batched Lipschitz bandit is reasonably interesting, although its applicability of the proposed algorithms is somewhat limited by the 'weak' regret bounds that are usual for Lipschitz bandit problems that have no further assumptions. In fact the low number of batches required (log log T) to achieve log-optimal regret is indicative of how little adaptivity is required in selecting the evaluation points.

For the problem in case, the authors do well to design an algorithm with low communication, and prove matching upper and lower bounds. The algorithm has low (time/space) complexity as well, although this is usually not a major problem for most Lipschitz bandit algorithms.

While the empirical work is not extensive, it is sufficiently illustrative to understand the performance of the algorithm. It would have been interesting to see some examples where the optimality gap between local optima is smaller.

The paper is well written with a clear structure.

While the relevant literature is discussed w.r.t. the proposed problem and results, there could be some deeper insight provided on how the proof techniques relate to the proof techniques used for similar problems.

---

> ### Author Response · Authors · 2022-08-02
> **Author Response**
>
> We would like to thank the reviewer for the effort reviewing our submission and the suggestions for improvement. We explain the reviewer's concerns point by point.
> >The algorithm has low (time/space) complexity as well, although this is usually not a major problem for most Lipschitz bandit algorithms.
>
> We agree that regret performance is most important for bandit algorithms. However, computational complexity is also relevant, especially for bandits in continuous space. As an instance, see Section 4 in [1].
>
> >How would be the bounds affected with a different Lipschitz constant, or if the constant would be unknown?
>
> The Lipschitz constant is known in most previous works of Lipschitz bandits, and often set to $1$ without loss of generality [2]. Strictly speaking, the Lipschitz constant for the reward functions affect the constant term in the regret bound.
>
> Under stronger assumptions, some previous works (e.g., [3]) consider Lipschitz bandits without knowing the Lipschitz constant. In [3], the reward function is assumed to be twice differentiable and have bounded Hessian. This assumption is stronger than Lipschitz continuity. Yet we think the setting of unknown Lipschitz constant is important, and will consider it in future works.
>
> >While there is some indication in e.g. line 85 of what the authors mean by optimality, it would be useful to specify it more formally, since it is referred in some theoretical statements (e.g. Corollary 2).
>
> Thank you for this suggestion. Indeed, `optimally solve' in Corollary 1 and 2 means that the algorithm reaches the optimal regret rate. We have clarified it in the revised version.
>
> >The lower bound for the adaptive case relies on the choice of number of batches $(B)$. This would exclude algorithms that adapts the number of batches depending on feedback. This is not a major limitation, and the proofs might be adapted to include this more adaptive variant, but the limitation could be mentioned by the authors.
>
> Thanks for the comment. Within our current framework, the complexity analysis (Theorem 5 and 6) shows that any algorithm (even the number of batches can be decided adaptively) cannot solve all problem instances using less than $\mathcal{O}(\log\log T)$ batches, if its regret rate is optimal. We have clarified this in the revised version.
>
> Yet we think that the adaptive determination of $B$ is a situation worth investigating in some other settings. We will consider it in future works.
>
> [1] Sébastien Bubeck, Rémi Munos, Gilles Stoltz, and Csaba Szepesvári. $\mathcal{X}$-armed bandits.
> [2] Robert Kleinberg, Aleksandrs Slivkins, and Eli Upfal. Multi-armed bandits in metric spaces.
> [3] Sébastien Bubeck, Gilles Stoltz, and Jia Yuan Yu. Lipschitz bandits without the Lipschitz constant.

---

> > ### Comment · Reviewer_ADGD · 2022-08-07
> > **Unkown Lipschitz constant**
> >
> > I agree that many papers assume known constant (including setting 1), but there are papers as well that discuss in passing the unkown case with a weaker bound (e.g., because the discretization is chosen without the knowledge of the constant). As the authors mentioned, there are papers that tackle the issue in a more principled way (DIRECT is probably one of the early solutions), and most of these papers indeed have some additional conditions.
> >
> > Nevertheless, I think, knowledge of the constant is unrealistic, and it is important to know the loss in performance/guarantee when the constant is not known.

---

> > > ### Author Response · Authors · 2022-08-09
> > > **Author Response**
> > >
> > > Thank you for your response. We agree with your comment, and will consider it in our further work.

---

### Official Review · Reviewer_fkQo · 2022-07-13

**Rating:** 6
**Confidence:** 4
**Soundness:** 3 good
**Presentation:** 3 good
**Contribution:** 3 good

**Summary:**

This paper studied the problem of Lipschitz bandit learning with batched feedback, where the expected reward is Lipschitz and the reward observations are communicated to the player in batches. The authors introduced a landscape-aware algorithm called BLiN to solve this problem. The authors showed that for a $T$-step problem with order-optimal regret with order-optimal batches (i.e., communication rounds).

**Questions:**


- Although the batched version of Lipschitz bandit learning is new, the analysis techniques used in proving the key results in Theorem 4 appear to be standard. It's unclear what new technical contributions this paper provides in the proof and it would be great if the authors could provide further clarifications.

- The experiments in this paper are rather basic and small-scale (two-dimensional). This paper could benefit from more extensive and larger-scale experiments to verify the theoretical results.

- There seems to be a mismatch between the motivating examples and the studied batched Lipschitz bandit problem in this paper. For example, in distributed computing, it is more typical that the batches are generated in a fixed periodic fashion. However, in later settings in this paper, the grid points (i.e., the timing of batched observations) are adaptive and can be controlled by the player. Could the author also analyze the periodic version of batched Lipschitz banding learning?

**Limitations:**

This paper is theoretical and not relevant to negative societal impacts.

**Strengths And Weaknesses:**

Strengths:

+ This paper studied a batch-version of Lipschitz bandit learning, which is new in the literature.
+ The authors established the order-optimality of communication for their proposed BLiN algorithm.

Weaknesses:

- Novelty and contributions are unclear in this paper.

- Experiments are rather limited.

---

> ### Author Response · Authors · 2022-08-02
> **Author Response**
>
> We would like to thank the reviewer for the effort reviewing our paper and the constructive suggestions. We explain the reviewer's concerns point by point.
> >Although the batched version of Lipschitz bandit learning is new, the analysis techniques used in proving the key results in Theorem 4 appear to be standard. It's unclear what new technical contributions this paper provides in the proof and it would be great if the authors could provide further clarifications.
>
> There are three main technical contributions in the paper.
> 1. Our work is the first to use elimination based method to solve Lipschitz bandits, which significantly reduces time and space complexity, and the number of communications.
> 2. We give the ACE sequence to get the optimal $\mathcal{O}(\log\log T)$ communication bound. In Lipschitz bandits literature, the problem of constructing such sequences has never been studied.
> 3. For our lower bound, the construction of hard instance for batched Lipschitz bandit problems is novel.
>
> >The experiments in this paper are rather basic and small-scale (two-dimensional). This paper could benefit from more extensive and larger-scale experiments to verify the theoretical results.
>
> The current work mainly presents theoretical contributions. We will include more experiments on harder cases and real-world datasets in future works.
>
> >There seems to be a mismatch between the motivating examples and the studied batched Lipschitz bandit problem in this paper. For example, in distributed computing, it is more typical that the batches are generated in a fixed periodic fashion. However, in later settings in this paper, the grid points (i.e., the timing of batched observations) are adaptive and can be controlled by the player. Could the author also analyze the periodic version of batched Lipschitz banding learning?
>
> Thank you for this suggestion. The batched bandits have many practical applications. We have replaced this example with one that is more relevant to active learning.
>
> To the best of our knowledge, periodic version has not been studied in batched bandits literature. We believe this is an interesting topic, because it is consistent with many real-world problems, as is mentioned by the reviewer. We will consider this setting in the future.

---

### Meta-Review · Area_Chair_duW5 · 2022-08-24

**Recommendation:** Accept
**Confidence:** Certain

**Metareview:**

This is an interesting work that initiates the study of Lipschitz bandits in the case of batched feedback. That it is possible to obtain nearly optimal regret with such little adaptation (only $\log \log T$ batches), as the authors do, is an interesting result; moreover, the regret lower bound based on the number of batches shows that the optimal regret can be achieved only with $\Omega(\log \log T)$ batches, implying the authors’ method A-BLiN has communication complexity no higher than methods obtaining the optimal regret.

All reviewers are positive on this work and the work merits acceptance; that said, as some reviewers suggested, the authors would do well to highlight the novelty of their approach and relate their proof techniques to proof techniques used for similar problems. Also, I noticed that the claims need to be adjusted somewhat. At times the authors claim that their regret rate is optimal or (prior to Definition 1 and in reference to A-BLiN) "without incurring increasing regret"; this is not correct, due to the extra $\log \log T$ factor. The authors should instead say that the regret rate is near-optimal and that the regret does not increase by much (but the rate certainly does worsen, however so slightly). It is important to be clear on optimality vs near-optimality. In any case, congratulations.

**Award:**

No

---

### Decision · Program_Chairs · 2022-09-14

Accept